# Enhancing the forecast accuracy of the daily number of patients arrivals in emergency department by hybrid ARIMAX-ANN algorithm

Hamed Tabesh[1], Ali AbbaszadehMozaffari[1], Zahra Ebnehoseini[2], Azadeh Saki[1]*

**1** Department of Epidemiology and Biostatistics, School of Health, Mashhad University of Medical Sciences, Mashhad, Iran, **2** Psychiatry and Behavioral Sciences Research Center, Mashhad University of Medical Sciences, Mashhad, Iran

* sakia@mums.ac.ir, azadehsaki@gmail.com

## Abstract

Accurate forecasting of daily arrivals in Emergency Departments (ED) is crucial for healthcare providers. This study incorporates a variety of factors, including meteorological and calendar influences, into the forecasting of ED patient arrivals. Due to the complex linear and nonlinear associations these factors have with ED arrivals, two hybrid algorithms were developing to enhance forecasting accuracy. These algorithms combine Auto-Regressive Integrated Moving Average (ARIMA) with Artificial Neural Network (ANN) methodologies, leveraging their strengths in handling linear and nonlinear relationships in patient arrival data. The first hybrid algorithm utilizes an ANN model with inputs comprising ARIMA-fitted values of the time series, ANN-fitted values of the ARIMA residuals, and ANN-fitted values of the time series, excluding nonlinear input features from ARIMA. The second hybrid approach combines ARIMA and ANN forecast values of the time series. Robust performance metrics were employ to validate the effectiveness of these hybrid algorithms, allowing for a clear comparison with standalone ARIMA, ANN, LSTM, and GLM forecasts at short, intermediate, long, and overall horizons. The overall accuracy indices for ARIMA, ANN, hybrid1, hybrid2, LSTM and GLM algorithms are RMSE = 37.43, 44.33, 39.33, 33.82, 55.53 and 34.96; ME = −16.38, 22.08, 13.93, 2.85, −36.24 and 5.65; SMAPE = 5.32, 6.57, 6.00, 5.18, 8.27, and 4.62, respectively. The hybrid2 algorithm demonstrates superior performance across short, intermediate, and overall horizons, while the ARIMAX model excels in long horizons characterized by low volatility. Although hybrid1 and ANN exhibit similar overall accuracy, the hybrid2 algorithm enhances accuracy indices specifically in the intermediate horizon, which is marked by high volatility. These findings significantly improve the predictive capabilities of existing algorithms and provide valuable insights for strategic decision-making in managing patient flow in emergency departments.

**Data availability statement:** The integrated time series of ED arrivals prepared for the current study are available at the S2 File.

**Funding:** The author(s) received no specific funding for this work.

**Competing interests:** The authors declare that they have no competing interests.

## Introduction

Accurate forecasting of daily patient arrivals in Emergency Departments (EDs) is crucial for optimizing resource allocation, enhancing patient care, and improving operational efficiency. As healthcare systems face increasing demands, understanding patient flow becomes essential for delivering high-quality services. Effective forecasting allows EDs to anticipate peak periods, manage staffing needs, and ensure the availability of necessary medical supplies. By utilizing historical data, statistical algorithms, and machine learning techniques, healthcare providers can predict patient volumes with greater precision.

As the healthcare landscape continues to evolve, the importance of accurate forecasting in EDs remains paramount, making it a critical focus for healthcare administrators and policymakers. Data-driven forecasts provide valuable insights that empower decision-makers to optimize staffing levels, budget allocations, and infrastructure improvements. Analyzing historical data also aids in developing long-term strategies for capacity planning, facility expansion, and service offerings based on projected demand trends.

A variety of methods applied to forecast ED patient arrivals. Regression-based prediction methods identify correlations between outcomes and input features, such as calendar or meteorological factors. Time series-based methods treat patient numbers as a time series, forecasting future values based on past data. Additionally, AI-driven forecasting techniques, including artificial neural networks (ANNs), support vector machines, decision trees, and Bayesian networks, commonly employed.

Recent literature highlights the development and evaluation of various hybrid algorithms for forecasting in time series analysis, particularly in contexts where data exhibits both linear and nonlinear patterns. These hybrid approaches aim to enhance forecasting accuracy by integrating multiple modeling techniques.

Zhang (2003) introduced a hybrid ARIMA–ANN algorithm, positing that a time series can be decomposed into linear and nonlinear components. The ARIMAX model captures the linear part, while ANN models the residuals to account for nonlinearity, yielding forecasts as the sum of both components [1]. Babu et al. (2014) proposed an MA-filter-based hybrid ARIMA–ANN algorithm, which improves upon Zhang's approach by decomposing time series into low and high volatility components, demonstrating superior accuracy [2]. However, Zhang's model's assumption of additive relationships may overlook interactions between components, potentially compromising performance [3].

Luo et al. (2017) developed a combinational forecasting algorithm for hospital outpatient visits, integrating seasonal ARIMA and exponential smoothing, which proved efficient for short-term forecasts [4]. Jilani et al. (2019) created a forecasting tool for UK emergency department (ED) attendance using historical data, comparing fuzzy time series with other algorithms and achieving acceptable accuracy for long-term predictions [5]. Yucesan et al. (2018) evaluated multiple methods, including ARIMA and ANN, for forecasting patient arrivals, finding that the ARIMA-ANN hybrid provided the best accuracy [6].

Erkamp et al. (2021) utilized a linear algorithm with feature selection to predict daily ED arrivals, omitting meteorological features due to their inaccuracy [7]. Vollmer et al. (2021) explored a predictive framework for hospital demand, revealing that generalized linear models often outperformed complex ensemble methods [8]. Rocha et al. (2021) tested various forecasting methods, concluding that recurrent neural networks and XGBoost yielded the most accurate forecasts [9]. Zhang et al. (2022) combined calendar and meteorological data with machine learning techniques to forecast patient arrivals, finding LSTM models superior for hourly predictions [10].

Tuominen et al. (2022) analyzed daily arrivals at a university hospital, employing advanced feature selection techniques and demonstrating that high-dimensional approaches can enhance predictive accuracy [11]. Gafni-Pappas et al. (2023) investigated forecasting methods for ED admissions across Spanish hospitals, finding ensemble methods to be particularly effective [12]. Iftikhar et al. (2023) developed a hybrid model for predicting mpox cases, showcasing its applicability to other infectious diseases [13]. Silva et al. (2023) reviewed predictive algorithms for ED visits, noting that while linear models were effective for short-term forecasts, machine learning methods offered greater stability across multiple horizons [14]. Jiang et al. (2023) conducted a meta-analysis of 35 articles, emphasizing the increasing use of AI-based algorithms and the importance of local calibration to improve forecasting accuracy in different ED contexts [15].

Overall, there has been a significant rise in research focusing on hybrid algorithms leveraging Artificial Intelligence for improving patient arrival forecasts in EDs [16]. Studies indicate that calendar features significantly influence ED arrivals, while meteorological factors may introduce uncertainty but are still relevant in some contexts.

While previous research has laid a strong foundation for time series forecasting in healthcare, significant limitations remain. When working with time series of ED arrivals, the problem of volatility, or heterogeneity in the variance of observations over time, is very common. Every explosion in the response has a cause. If we do not have a predictor variable to determine the explosion in the data, ARCH or GARCH models are suitable, but these models do not have long-term predictive power and tend towards the mean without any dispersion over intermediate or long-term horizons. Calendar events and meteorological conditions are significant drivers of surges in emergency department (ED) arrivals in Iran. Seasonal changes, holidays, and local festivities often lead to increased activity in healthcare facilities as people engage in various social gatherings and outdoor activities. For instance, during national holidays or religious observances, there tends to be a higher incidence of accidents and health-related issues, resulting in a spike in ED visits.

Additionally, meteorological factors such as extreme temperatures, rainfall, and air quality can profoundly impact public health. For example, during heatwaves, there is often an increase in heat-related illnesses, while heavy rains can lead to accidents or exacerbate respiratory conditions due to poor air quality.

Moreover, studies have shown that specific weather patterns, such as high humidity or sudden temperature fluctuations, can correlate with increased hospital admissions, particularly for vulnerable populations such as the elderly or those with pre-existing health conditions.

Understanding these dynamics is crucial for healthcare planning and resource allocation. By analyzing historical data on ED visits in relation to calendar events and weather conditions, healthcare providers can better anticipate patient influxes and ensure that adequate staff and resources are available to meet demand. This proactive approach can ultimately enhance patient care and improve outcomes in emergency situations.

The contribution of this study is centered on a systematic empirical evaluation of established forecasting approaches within a highly variable emergency department (ED) setting. Using data from Mashhad, Iran, the study examines how ARIMA-based, machine learning, and hybrid ARIMA–ANN configurations perform under conditions strongly influenced by calendar effects, religious events, tourism, and meteorological variability. A key aspect of the proposed framework is the integration of calendar and weather-related variables selected through the Maximal Information Coefficient (MIC), combined with multiple-testing–adjusted significance measures, to capture both linear and nonlinear associations. By assessing forecast accuracy and stability across short-, intermediate-, and long-term horizons, the study provides practical

insights into how commonly used hybrid strategies can be adapted to support ED staffing and operational planning in environments characterized by extreme demand fluctuations.

To achieve precise forecasts, the study proposes a combination of ARIMA and ANN algorithms. Initially, the optimal ARIMAX model for linear relationships is identified using metrics such as Mean Square Error (MSE), Akaike Information Criterion (AIC), and Bayesian Information Criterion (BIC). Subsequently, the best ANN algorithm for capturing nonlinear relationships is determined based on the MSE index. The analysis reveals patterns of overestimation in ANN forecasts and underestimation in ARIMA forecasts across different time horizons.

Two hybrid algorithms are then developed to address the trade-offs between overestimations and underestimations across various time horizons. The first hybrid algorithm utilizes ANN inputs, which include ARIMA-fitted values, ANN-fitted values of the ARIMA residuals, and ANN-fitted values of the time series, while excluding certain nonlinear input features. The second hybrid approach combines ARIMA and ANN forecast values of the time series.

In addition to statistical, machine learning, and hybrid ARIMA–ANN approaches, a Long Short-Term Memory (LSTM) model is included as an exploratory baseline to provide a reference point for comparison with more established forecasting frameworks. The LSTM is not intended as a primary modeling contribution, but rather as a benchmark representing a widely used deep learning approach for time-series prediction.

Finally, the forecasting accuracies for daily ED patient arrivals across short, intermediate, and long horizons are compared between the hybrid algorithms and standalone ANN, ARIMAX, LSTM, and GLM algorithms, employing various accuracy metrics, including RMSE, MPE, MAE, MAPE, SMAPE, ME, and R-square. This comprehensive approach aims to enhance the accuracy of patient arrival forecasts, ultimately improving operational efficiency in emergency departments.

## Materials and methods

### Data and features

To demonstrate the efficiency of proposed hybrid algorithms, this study utilized the dataset sourced from the Health Information System (HIS) database of ED of Imam Reza hospital in Mashhad, a heavily burdened referral hospital for emergency cases in Northeast Iran. The dataset includes the daily count of patients referred to the emergency department over a span of three Iranian official calendar years: 1395, 1396, and 1397, corresponding to the period from 21 March 2016–18 March 2019. The authors had not access to information that could identify individual participants during or after data collection. Ethics of this Study approved at Research Ethics Committee of Mashhad University of Medical Sciences with the reference number: IR.MUMS.REC.1399.443.

Calendar features were from Iran's official calendars, while meteorological features obtained from the website of the Meteorological Organization at https://data.irimo.ir/. The output was the daily number of patients referred to the emergency department, and the input features comprised calendar and meteorological attributes for each day. Calendar features encompassed year, month, day of the week, holidays and post-holiday periods, and calendar events. Daily meteorological features included average, minimum, and maximum temperature, average and maximum wind speed, average rainfall, average relative humidity, and sunny hours. There are no missing values in the output, the ED arrival data, or any of the input features.

Model training utilized the initial 1008 days (equivalent to 144 weeks) of available data. To evaluate the robustness of algorithms and forecasting accuracies, the remaining 84 days (12 weeks) were held out for testing. This standardized train/test partition was maintained uniformly across the evaluation of all algorithms: Artificial Neural Network (ANN), ARIMAX, proposed Hybrid models, Long Short-Term Memory (LSTM), and Generalized Linear Model (GLM).

### Feature selection approach

To explore the association between input features and the number of arrivals in ED the Maximal Information Coefficient (MIC) used. The application of MIC for feature selection had a significant advancement. This nonparametric method allows for a comprehensive exploration of the relationships between input features and patient arrivals, accommodating

nonlinear associations that traditional correlation methods might miss. Since the relation between the arrival of ED patients and input features may be nonlinear, it is not appropriate to use the Pearson correlation coefficient to explore the relationships [17]. The Maximum Information Coefficient (MIC) is utilized to identify the most relevant features in the ARIMA algorithms by assessing the strength of the relationships between meteorological and calendar features and the number of patient arrivals. Since many significant input features are highly correlated, it is not feasible to include all of them in the ARIMA model. Therefore, the input features selected in a forward manner based on their MIC values and adjusted p-values.

The MICs and their p-values calculated by "minerva" package and the adjusted p-values for multiplicity by two multiplicity adjustment procedures calculated with coding in R4.4.2. The first is the Benjamini-Hoschberg FDR (BH-FDR) procedure, which assumes that the input features are independent. The second is the Modified FDR (M-FDR) procedure, which modified the BH-FDR procedure based on information theory for dependent features [18]. Since the input features in the present study are dependent, the second procedure was used for feature selection in the ARIMAX and GLM models, enabling a comparison of model performance with and without the Maximal Information Coefficient (MIC) in forward feature selection.

## The ARIMA/ARIMAX algorithm

The Autoregressive Integrated Moving Average (ARIMA) model is a popular and powerful time series forecasting technique that can used to predict future values based on past observations. The ARIMAX (p, d, q, b) algorithm is a type of autoregressive integrated moving average (ARIMA) time series algorithm which include exogenous input features in decision-making and forecasting. The notation ARIMAX (p, d, q, b) refers to the specific characteristics of the algorithm:

Autoregressive (AR) terms: The parameter p represents the number of autoregressive terms in the algorithm. Autoregressive terms capture the relationship between an observation and a certain number of lagged observations.

Integrated (I) terms: The parameter d indicates the number of differences required to make a time series stationary in an integrated series.

Moving Average (MA) terms: The parameter q represents the number of moving average terms in the algorithm. Moving average terms capture the relationship between an observation and a random error term from a previous period.

Exogenous inputs (X): The parameter b represents the number of exogenous input terms in the algorithm. These are exogenous features where excluded of ARIMAX but are included as additional predictors in the algorithm.

$$\nabla^d y_t - m_t = \varepsilon_t + \sum_{i=1}^{P} \varphi_i \left( y_{t-i} - m_{t-i} \right) + \sum_{j=1}^{q} \theta_j \varepsilon_{t-j}$$

$$m_t = c + \sum_{i=1}^{b} \eta_i x_{it}$$

Where $y_t$ is the number of ED patients at time t, $\nabla^d$ is the differences of order d; which defined as

$$\nabla^d = (1 - B)^d$$

B is the backshift operator;

$$By_t = y_{t-1}, \; and \; B^k y_t = y_{t-k}$$

for instance:

$$\nabla^2 y_t = (1 - B)^2 y_t = \left(1 - 2B + B^2\right) y_t = y_t - 2y_{t-1} + y_{t-2}$$

$x_{it}$ incorporates $i^{th}$ exogenous (input) feature, $c$ is an unknown constant term, $\eta_i$ is the coefficient parameter of $i^{th}$ input feature, $\varepsilon_t$ is independent random variables distributed as normal with mean 0 and variance $\sigma^2$, $\varphi_i$ (i = 0, 1,... p) are the algorithm parameters of Auto-Regressive (AR) part, and $\theta_j$ (j = 0, 1,...,q) are the algorithm parameters of Moving-Average(MA) part.

The ARIMAX fitting process involves several key steps. First, the stationarity condition checked using the Augmented Dickey-Fuller Test, where a p-value less than the significance level indicates stationarity. Next, the ARIMA parameters is determined using the Partial Autocorrelation Function (PACF) to identify the order of the autoregressive (AR), the Autocorrelation Function (ACF) to identify the order of the moving average (MA), and any required differencing. The AIC and BIC criteria are then used to find the optimal parameters for the AR, differencing (I), MA, and the number of exogenous input terms in the algorithm, with lower AIC/BIC values indicating a better algorithm fit. Finally, the forecast accuracy evaluated using three diagnostic statistics: Root Mean Squared Error (RMSE), Mean Error (ME), Mean Absolute Error (MAE), Mean Percentage Error (MPE), and Mean Absolute Percentage Error (MAPE), which are used to compare the forecast performance across different algorithms.

## ANN algorithm

Artificial neural network (ANN) is one of the most widely used algorithms in approximating functions and predictions that has successful application in many areas especially time series algorithms. One of the greatest advantages of an ANN is its ability to approximate complex nonlinear relationship without prior assumptions about the nature of the data structure. The mathematical relationships between the output $Y_t$ and the inputs lagged values of the target feature or lagged values of the input features ($X_{t-1}, \ldots, X_{t-p}$) can expressed as follows:

$$\hat{y}_t = h(x) = \sum_{j=1}^{q} w_j . g \left( b_i + \sum_{i=1}^{p} w_{ij} x_{t-i} \right)$$

Where $w_j$ are the weights connecting the hidden layer neurons to the output layer neuron, g is an activation function that introduces non-linearity into the algorithm, $b_i$ represents the bias term associated with the hidden layer neuron i, $w_{ij}$ are the weights connecting the lagged input features of time t to the hidden layer neuron i, $x_{t-j}$ represents the lagged input features of time t, $p$ and $q$ are the number of input and hidden nodes respectively.

The formula calculates the weighted sum of the lagged inputs at time t, applies the activation function, and then computes the final output using the weights and biases. By including lagged values of the target feature or input features, the algorithm can capture the temporal dependencies in the time series data.

Typically, activation function is a sigmoidal or the hyperbolic tangent. These functions introduce non-linearity into the network, allowing it to learn and approximate complex non-linear relationships within the data. Sigmoid functions are smooth and continuously differentiable, making well-suited optimization algorithms. This function allows for efficient learning and updating of the network weights through gradient descent methods. Therefore, the sigmoid function used for activation function that shown as:

$$g \left( b_i + \sum_{i=1}^{p} w_{ij} x_{t-i} \right) = \frac{1}{1 + \exp(b_i + \sum_{i=1}^{p} w_{ij} x_{t-i})}$$

Hence, the ANN algorithm fits a nonlinear functional mapping from the input features $(X_{t-1}, \ldots, X_{t-p})$ to the future value $y_t$.

In ANN, one of the important problems is to select an appropriate number of hidden nodes $q$ and the dimension of the input vector. In practice, experiments are often conducted to select the appropriate values $p$ and $q$ by trial and error approach. After the network structure $(p, q)$ is specified, then the parameters will be adjusted by some learning rules. Among the several learning algorithms available, the back-propagation (BP) used in this study.

**ANN-ELM algorithm**

Extreme Learning Machine (ELM) for time series forecasting is a fast and efficient neural network-based approach that leverages a single-layer feedforward neural network (SLFN) structure. The key idea behind ELM is that the input weights and hidden layer biases are randomly assigned and remain fixed during training, while only the output weights are learned, typically by solving a linear system using matrix pseudo-inversion.

**Gradient function for feature importance in ANN**

the gradient function approach for feature importance in Artificial Neural Networks (ANNs) is a powerful, model-specific method to understand which input features most influence the network's output.

In an ANN, the gradient measures how sensitive the model's output is to changes in each input feature. Mathematically, it's the partial derivative of the loss function (L) with respect to each input variable $x_i$:

$$\frac{\partial L}{\partial x_i}$$

The derivative quantifies both the magnitude and direction of change in the model output resulting from infinitesimal variations in the input feature $x_i$. Features exhibiting larger absolute gradient values are deemed more influential, as minor perturbations in these inputs induce greater changes in the output. To ensure comparability, the raw gradient magnitudes are generally normalized across all features, yielding their relative percentage contributions to the model prediction.

**First proposed hybrid algorithm**

Following ARIMAX and ANN algorithms the hybrid1 algorithm conducted based on the outputs of ARIMAX and ANN. Here is the steps of the hibrid1 algorithm.

A given time series $y_t$ considered as a function of the linear and nonlinear component as below:

$$y_t = f(L_t, N_t^1, N_t^2, X^c)$$

Where $L_t$ is the linear component, representing the fitted values from the ARIMAX, $N_t^1$ is the first nonlinear component, estimated by fitting an ANN to the residuals of the ARIMAX, $N_t^2$ is the second nonlinear component, derived from the fitted values of the ANN, $X^c$ is the matrix of remaining exogenous input features that were not included in the ARIMAX.

The proposed algorithm involves the following steps:

1. Estimating the linear component $L_t$

Capture the linear component $L_t$ by fitting the ARIMAX. This step aims to identify the linear patterns in the data and obtain the fitted values and residuals for use in subsequent steps.

2. Estimating the nonlinear component $N_t^1$

Apply an ANN to the residuals of the ARIMAX. This step captures any nonlinear relationships that may exist and estimates the first nonlinear component $N_t^1$.

3. Estimating the nonlinear component $N_t^2$

Fit an ANN to capture the second nonlinear component $N_t^2$. This step focuses on modeling the nonlinear patterns through ANN and obtaining the fitted values for use in the final step.

   4. Combine components and exogenous features

Fit an ANN that combines the fitted values from the previous steps $L_t$, $N_t^1$, and $N_t^2$ along with the remaining exogenous input features $X^c$ that were excluded from the ARIMAX.

## Second proposed hybrid algorithm

This proposed hybrid algorithm based on the assumption that time series data may be linear and nonlinear components, therefore the forecasts could estimate by combining both linear and nonlinear forecast at time horizon h as follows:

$$y_{t+h} = f(L_{t+h}, N_{t+h})$$

Where:

   $L_t$ Represents the linear component, estimated using the ARIMAX.

   $N_t$ Represents the nonlinear component, estimated using an ANN.

   Steps of the Hybrid Algorithm:

   Step1: Estimating the linear component $L_{t+h}$

   The ARIMAX fitted to the given time series $y_t$ to capture the linear patterns. The corresponding linear forecasts at the time horizon h $\hat{L}_{t+h}$ generated from this algorithm.

   Step2: Estimating the nonlinear component $N_{t+h}$

   An ANN fitted to the same time series $y_t$ to capture the nonlinear relationships that linear algorithms cannot adequately do. This results in the estimated nonlinear component forecast at the time horizon h, $\hat{N}_{t+h}$.

   Step3: Estimating the forecasts $y_{t+h}$

   The combined forecast at the time horizon h, $\hat{y}_{t+h}$ can obtained using the following equation:

$$\hat{y}_{t+h} = (\hat{L}_{t+h} + \hat{N}_{t+h})/2$$

This equation calculates the mean of the forecasts from both the linear and nonlinear components, providing a more robust forecast for the time series.

   Fig 1, shows the schematic diagram of both Hybrid1 and Hybrid2, algorithms.

## GLM

Generalized Linear Models (GLMs) are a flexible framework for modeling various types of response variables, including those encountered in time series analysis. They extend traditional linear regression by allowing for response variables that have error distribution models other than a normal distribution. The response variable Y is assumed to follow a probability distribution from the exponential family (e.g., normal, binomial, Poisson, Negative Binomial). This allows GLMs to handle various types of data, such as counts, proportions, and continuous measurements. The relation between input features and a specific function of mean response (link function) is linear. Which allows the nonlinear regression between input features and outcome.

   As the daily number of ED arrivals, $Y_t$ is a count response Poisson or negative binomial distributions are appropriate. So, the canonical link function is log and the model is,

$$log(E\left(Y_t|x_t\right)) = \beta_0 + \sum_{i=1}^{p} \beta_i x_{it}$$

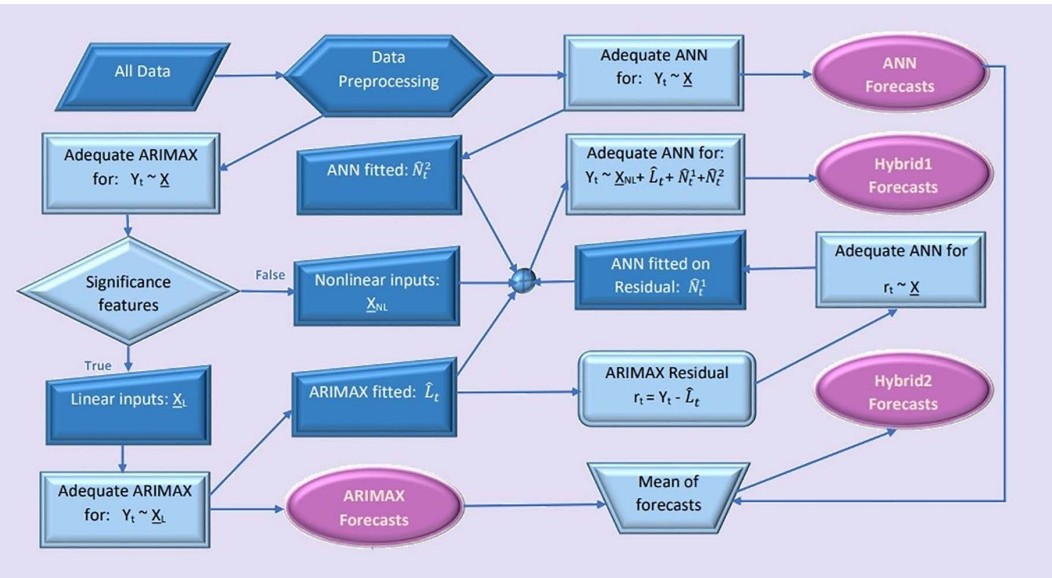

**Fig 1. Schematic flowchart of the proposed hybrid algorithms.**

Where $x_t$ is the set of p input features at time t, and $x_{it}$ is the i<sup>th</sup> input feature at time t. Also, the identity link function leads to ordinary regression:

$$E\left(Y_t \middle| x_t\right) = \beta_0 + \sum_{i=1}^{p} \beta_i x_{it}$$

However, the maximum likelihood estimations for the parameters are based on either Poisson or Negative Binomial distributions.

The choice between the Poisson and Negative Binomial distributions for count data depends on the significance of overdispersion. If overdispersion is significant, the Poisson distribution is not appropriate, and the Negative Binomial distribution should be used instead.

## LSTM

Long Short-Term Memory (LSTM) networks are a type of recurrent neural network (RNN) specifically designed to model sequential data. Unlike traditional RNNs, which struggle with long-range dependencies due to issues like vanishing and exploding gradients, LSTMs are capable of learning and retaining information over extended periods. This makes them particularly effective for tasks involving time series prediction, natural language processing, and other applications where context from previous inputs is crucial.

The LSTM architecture consists of a series of memory cells, each containing three primary components: input gate, forget gate, and output gate. These gates regulate the flow of information, allowing the network to learn when to remember or forget information.

Input Gate: This gate determines how much of the new input information should be stored in the cell state. It takes the current input and the previous hidden state as input and applies a sigmoid activation function to produce values between 0 and 1. A value of 0 means "ignore this input", while a value of 1 means "completely accept this input."

Input Gate Activation: $i_t = g(w_i \cdot [h_{t-1}, x_t] + b_i)$

Forget Gate: The forget gate decides what information from the previous cell state should be discarded. Similar to the input gate, it uses a sigmoid activation function to produce a value between 0 and 1 for each element in the cell state. This allows the model to forget irrelevant information selectively.

Forget Gate Activation: $f_t = g(w_f \cdot [h_{t-1}, x_t] + b_f)$

Output Gate: This gate controls what information from the cell state should be output to the next layer or time step. It uses both the current input and the previous hidden state to determine the output, applying a sigmoid function to filter the cell state.

Output Gate Activation: $o_t = g(w_o \cdot [h_{t-1}, x_t] + b_o)$

The cell state is the core component of the LSTM that carries information across time steps. It is updated at each time step based on the contributions from the input and forget gates. The update process can be described mathematically as follows

Candidate Cell State: $\widetilde{C}_t = g(w_c \cdot [h_{t-1}, x_t] + b_c$

Cell State Update: $C_t = f_t * C_{t-1} + i_t * \widetilde{C}_t$

Where, g is the sigmoid activation function, w represents the weight matrices, b represents the bias vectors for each gate, $h_t$ is the hidden state at time t, and $x_t$ is the set of input features at time t.

The LSTM hyperparameters—including the look-back window length, number of hidden units, batch size, and training epochs—were selected within commonly adopted ranges reported in the literature and refined through preliminary experiments aimed at reducing validation error, rather than through exhaustive hyperparameter optimization. In addition, a limited sensitivity assessment was carried out during these preliminary trials to confirm that the selected configuration provides stable forecasting performance across reasonable variations of the main hyperparameters.

## Accuracy indices

The forecast accuracies among ARIMAX, ANN, and hybrid algorithms compared using different Metric Distance Scaling (MDS) for estimating the dissimilarity between actual values and forecasting values. Suppose that $\delta_t$ is dissimilarity between actual and forecast values. It can be transformed to a distance with a monotonic function f, so:

$$d_t \approx f(\delta_t)$$

A metric Distance Scaling (MDS) consider

$$f(\delta_t) = \alpha + \beta \delta_t$$

Where:

$$\alpha = 0, \ \beta = 1 \ \rightarrow \ \textit{Absolute MDS}$$

$$\alpha = 0, \ \beta > 1 \ \rightarrow \ \textit{Ratio MDS}$$

$$\alpha \geq 0, \ \beta \geq 0 \ \rightarrow \ \textit{Interval MDS}$$

If $\delta_i$ is similarity rather than dissimilarity, then we need $\beta < 0$.

The present study employed both Absolute and Ratio forms of MDS, assessed using a suite of standard accuracy indices. These metrics included: Mean Error (ME), Root Mean Square Error (RMSE), Mean Absolute Error (MAE), Mean

Absolute Percentage Error (MAPE), Standardized Mean and Absolute Percentage Error (SMAPE). The definitions of all accuracy indices are provided below, where t denotes the forecasting day, $y_t$ = observed value on day t, $\hat{y}_t$ = forecast value on day t, and t = 1,2, …, T.

The ME measures the mean of the forecast errors;

$$\frac{1}{T}\sum_{t=1}^{T}(y_t - \hat{y}_t)$$

**2-**RMSE calculates the square root of the mean of the squared differences between forecasts and actual values;

$$\sqrt{\frac{1}{T}\sum_{t=1}^{T}(y_t - \hat{y}_t)^2}$$

3-MAE computes the mean of the absolute differences between forecasts and actual values;

$$\frac{1}{T}\sum_{t=1}^{T}|y_t - \hat{y}_t|$$

4-MPE measures the mean percentage difference between forecasts and actual values;

$$\frac{1}{T}\sum_{t=1}^{T}\frac{(y_t - \hat{y}_t)}{y_t} \times 100$$

5-MAPE measures the mean of the absolute percentage difference between forecasts and actual values;

$$\frac{1}{T}\sum_{t=1}^{T}\left|\frac{(y_t - \hat{y}_t)}{y_t}\right| \times 100$$

6-SMAPE measures the weighted (standardized) mean of the absolute percentage difference between forecasts and actual values;

$$\frac{1}{T}\sum_{t=1}^{T}\frac{|y_t - \hat{y}_t|}{(|y_t| + |\hat{y}_t|)/2} \times 100$$

7-R-Square measures the coefficient of determination of observed by forecasting values; The distance $d_t$ is fitted to the function $f(\delta_t)$ using a least-squares (LS) loss function:

$$L_f = \sum_{t=1}^{T}(d_t - f(\delta_t))^2$$

This approach is referred to as Metric Least-Squares Scaling, and its goodness of fit can be evaluated using the coefficient of determination (R-Square).

8-Diebold-Mariano test

The Diebold-Mariano (DM) test is a statistical test used to compare the predictive accuracy of forecasting models. It's particularly useful in time series analysis, where the goal is to determine the significance difference between actual values and forecasted values over a given period.

The test statistic for the DM test is calculated as follows:

$$DM = \frac{\bar{d}}{\frac{1}{T}\sum_{t=1}^{T}(d_t - \bar{d})^2}$$

Where $d_t$ is distance index, and $\bar{d}$, is the mean of $d_t$ for t = 1, 2, 3, …, T. If the calculated DM statistic exceeds the critical value from the standard normal distribution, it indicates that the forecast values differ significantly from the actual values.

Comparing these performance indices across different forecasting algorithms provides insights into the effectiveness of each approach for specific datasets. The measurement units of the time series influence the values of ME, RMSE, and MAE. While RMSE, MAE, MAPE, SMAPE and R-Square are always positive, ME and MPE can be positive or negative, indicating overestimation or underestimation in the forecast values, respectively. A near-zero ME or MPE does not necessarily indicate the best forecasting accuracy, as it may simply reflect a balance between positive and negative residuals. The MAPE, being unit-less, can used to compare the overall accuracy of different algorithms. The algorithm with the smallest MAPE value exhibits the best performance relative to the others. Evaluating all the indices is necessary to determine the strengths, weaknesses, and overall performance of the various forecasting algorithms.

The TSA, ANN, and hybrid TSA-ANN algorithms fitted to the ED patient arrivals data set by packages; "stats", "nnfor", "aTSA", and "forecast" in R4.3.2, LSTM algorithm fitted in Python. All the R and Python codes of this study presented in S1 File and the data set attached in S2 File of supplementary materials.

## Results

The Emergency Department (ED) at Imam Reza hospitals received 641511 patients during the 3-year study period. The variation in daily arrivals of ED patients is between 439 and 874 with an average of 585.32 ± 69.44 arrivals per day, and the median, first and third quartiles were 579, and (530, 633), respectively. The number of ED patients remained relatively consistent across the years 1395, 1396, and 1397, with 214119, 216698, and 210694 visits, respectively.

### Preliminary data analysis

Fig 2 shows the time series plot of ED patients (daily number of patient arrivals in ED) for three years of study. It shown in Fig 2 that at the Nowruz holidays, the first days of the New Year in the Iranian official calendar the ED patients have maximum and after these holidays the number of ED patients suddenly decreased and then gradually increased during the first semester of the year. With School openings at the beginning of the second semester of each year, the number of ED patients suddenly decreased again.

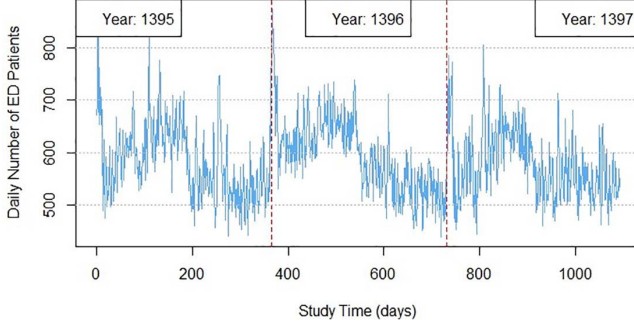

**Fig 2. Time series plot of the patients arrivals in ED.**

The data preprocessing on input features where included in these study categorized in MIC indices and graphical discovery. Table 1 summarizes the characteristics of the input features and presents their Maximal Information Coefficient (MIC) values, along with adjusted p-values from both the Benjamini-Hochberg (BH) and Modified False Discovery Rate (M-FDR) procedures, in relation to patient arrivals. The BH-FDR procedure identified all features as significant at $\alpha = 0.05$ level, with the exception of the year feature (X1). This suggests that most features meaningfully contribute to predicting patient arrivals. In contrast, the M-FDR procedure identified only 12 significant features, indicating that different statistical methods can yield varying results regarding the relevance of predictors. The MIC values indicate the strength of association between each feature and patient arrivals.

As shown in Table 1, significant calendar features include semester, month, and holiday status, which reflect temporal patterns in patient arrivals. For example, the semester and month features show MIC values of 0.37, indicating a robust association with patient arrivals. Daily meteorological features such as maximum temperature (X9) and average temperature (X11) also demonstrate strong associations, with MIC values of 0.37 and 0.38, respectively. These findings suggest that weather conditions significantly influence patient arrivals in emergency departments. The adjusted p-values from both

**Table 1. Characteristics of Input Features.**

| Input features | Feature No. | Label | Description | MIC | M-adj. p-value | BH-adj. p-value |
|---|---|---|---|---|---|---|
| Calendar Features | X1 | Year | Three years in Iranian official Calendar: 1395,1396,1397 | 0.08 | 1.00 | 0.10 |
| | X2 | Semester | Semester 1: Spring & Summer Semester 2: Fall & Winter | 0.37 | 0.00* | 0.00* |
| | X3 | Month of Year | 1:12, Based on Iranian official Calendar | 0.37 | 0.00* | 0.00* |
| | X4 | Day of Week | 1: Sat, 2: Sun, 3: Mon, 4: Tue, 5: Wen, 6: Tur, and 7: Fri (Weekend) | 0.1 | 0.14 | 0.01* |
| | X5 | Recode X4 | 0: Sun, Mon, and Tue 1: Tur, Fri, and Sat | 0.16 | 0.04* | 0.00* |
| | X6 | Holiday | 0: No Holiday 1: Holiday | 0.16 | 0.03* | 0.00* |
| | X7 | After Holiday | 0: No After Holiday 1: After Holiday | 0.11 | 0.51 | 0.03* |
| | X8 | Calendar Events | 0: No Event, 1: Islamic Traditions, 2: Ancient Traditions | 0.16 | 0.06 | 0.00* |
| Meteorological Features | X9 | Max. Temperature | Maximum Temperature (°C) in 24 Hours | 0.37 | 0.00* | 0.00* |
| | X10 | Min. Temperature | Minimum Temperature (°C) in 24 Hours | 0.35 | 0.00* | 0.00* |
| | X11 | Average Temperature | Mean of Temperatures (°C) in 24 Hours | 0.38 | 0.00* | 0.00* |
| | X12 | ΔTemperature | Difference between Maximum and Minimum Temperature in 24 Hours | 0.15 | 0.00* | 0.00* |
| | X13 | Max. Wind Speed | Maximum of Wind Speed (Km/h) in 24 Hours | 0.15 | 0.00* | 0.00* |
| | X14 | Average Wind Speed | Mean of Wind Speeds (km/h) in 24 Hours | 0.13 | 0.02* | 0.00* |
| | X15 | Average Raining | Mean of Raining (mm) in 24 Hours | 0.13 | 0.28 | 0.01* |
| | X16 | Average Humidity | Mean of Relative Humidity (%) in 24 Hours | 0.26 | 0.00* | 0.00* |
| | X17 | Sunny Hours | Sunny Hours in 24 Hours | 0.26 | 0.00* | 0.00* |

*Significance at $\alpha=0.05$.

BH and M-FDR procedures provide insights into the reliability of the identified features. Most features have significant adjusted p-values (<0.05), indicating strong statistical significance. By considering BF-FDR adjusted p-values except the year, all other 16 features have significance association with ED patient arrivals. While according to strong correlation between input features, M-FDR adjusted p-values detect 12 significant features.

Therefor Table 1 illustrates the importance of selecting relevant features for predictive modeling of patient arrivals. The use of MIC and adjusted p-values highlights the complex relationships between various calendar and meteorological factors, providing a comprehensive understanding of their impact on emergency department utilization.

Fig 3 illustrates the variability of emergency department (ED) patients according to calendar features. In the Iranian official calendar, the New Year coincides with the first day of spring, resulting in months that correspond to the seasonal changes of spring, summer, autumn, and winter. The data presented in Fig 3(1) indicates that the number of patients arriving at the ED is not constant throughout the months; at the first month of the year the median and the variation of ED patients arrival is maximum, and in following months the medians and variations decrease gradually. Fig 3(2) shows the number of ED patient's arrivals in the first half of the year are significantly higher than in the second half. Fig 3(3) is the box-plot of patient arrivals by weekdays reveals an increase during the weekends (Thursday and Friday) and on the first day of the week (Saturday). To better presenting this trend, a new feature categorizing weekdays into two groups created, Fig 3(4). Furthermore, Fig 3(5) and Fig 3(6) show that patient numbers rise on vacations and the days following vacations. In the context of the Iranian calendar, two types of traditional events—ancient and religious—are noteworthy. Fig 3(7) indicate that the number of ED patients is generally higher on ancient tradition days compared to religious tradition days. Fig 3(8) shows consistency in patient arrivals during three years.

Fig 4 expands the data preprocessing by depicting the variation in ED patients based on meteorological features. Fig 4(1) suggests a positive relation between average temperature and the number of ED patients; as temperatures rise, patient arrivals increase. However, the relationship between the number of ED patients and the temperature difference (Δt, defined as the maximum temperature minus the minimum temperature) is not monotonic, Fig 4 (2). Specifically, the number of patients increases with Δt up to 20°C, after which it begins to decline. Additionally, a similar trend observed with mean wind speed, where patient numbers increase with higher wind speeds up to 6 km/h and after which it declines Fig 4 (3). Conversely, Fig 4 (4) and Fig 4 (5) show a downward trend in the number of ED patients with increasing average rainfall and humidity over a 24-hour period. Finally, Fig 4 (6) indicates a consistent increase in patient arrivals with more hours of sunshine during the day.

## Building the ARIMAX

The Augmented Dickey-Fuller (ADF) Test showed a unit root for a raw response. After the first difference in raw response, the ADF statistics were significant with $p < 0.01$, and the stationary condition achieved. Fig 1 shows the time series plot of raw response (daily number of patient arrivals in ED). The ACF and PACF plots to evaluate the autocorrelations and seasonal trends shown in Fig 5.

The plots of sample ACF and PACF used for selecting the order of an ARIMA model. As shown in Fig 5(1), the time series has not stationary condition because the sample ACF series decays very slowly and has a weekly seasonal trend, the Fig 5(2), shows the first-order autocorrelation in raw data. Therefore, we differenced the original data once at first lag and the ACF and PACF plots shown in Figs 5(3) and 5(4). Based on these figures, following a first difference, the stationary condition achieved. The PACF indicates negligible seasonal autocorrelations that diminish at lags that are multiples of seven. Since the day of the week was an input feature, this seasonal pattern captured by this feature, negating the need for a seasonal order in the ARIMAX. After stepwise feature selection, only six significant features (X2, X5, X6, X7, X8, X11) remained in the final ARIMAX model due to dependency among input features. The results of the ARIMAX shows in Table 2.

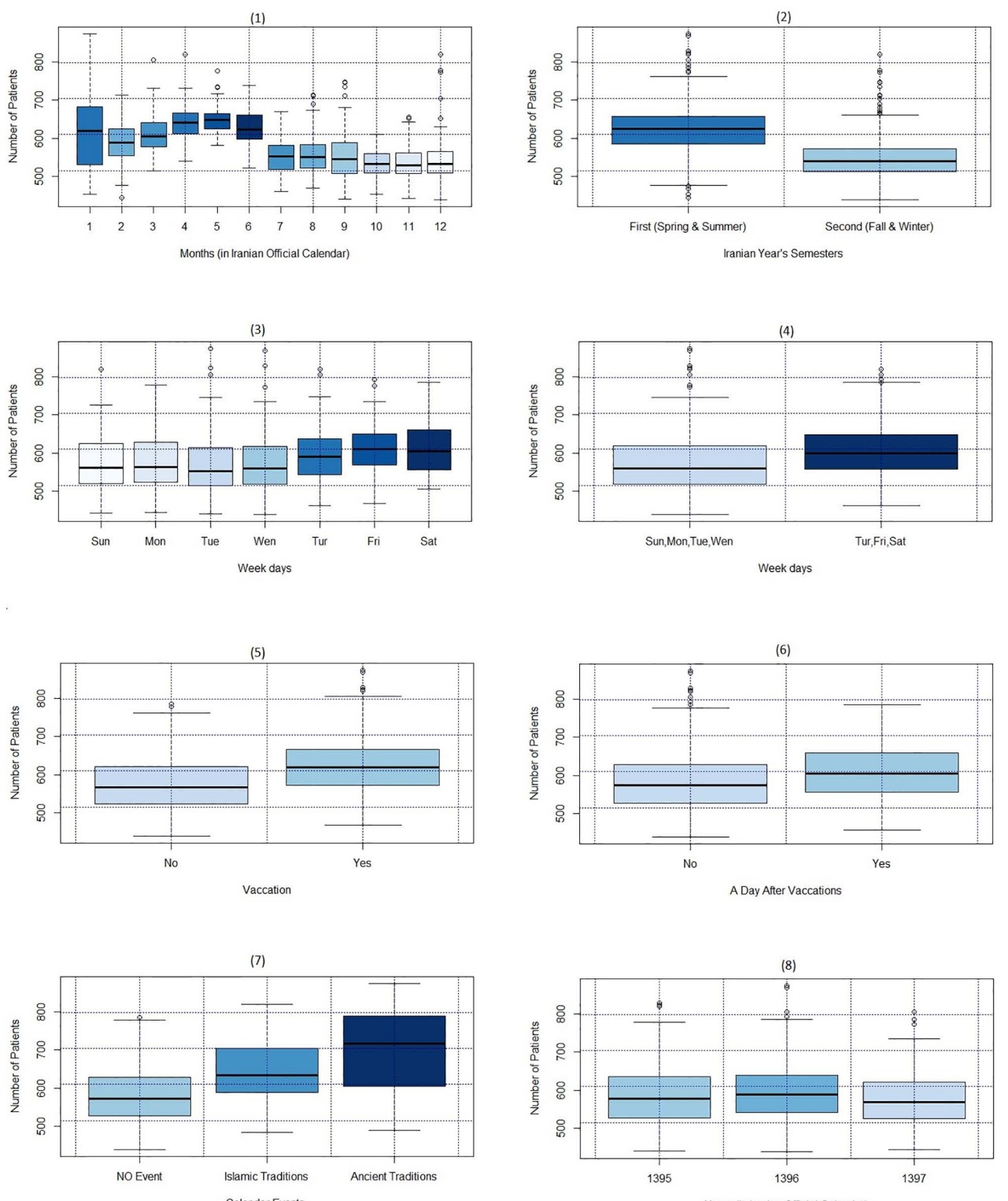

**Fig 3. Box-plot of the ED patients by calendar features.**

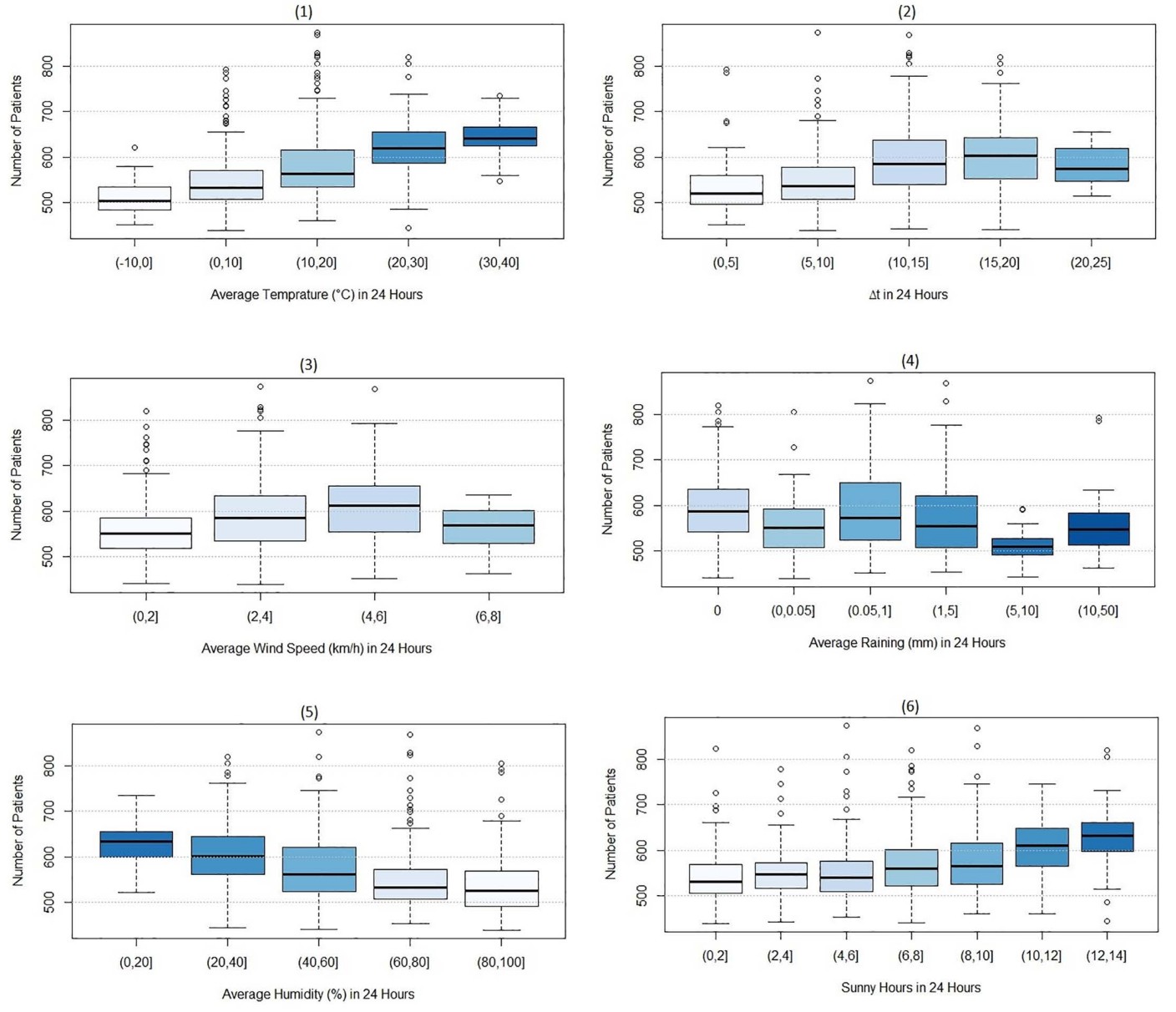

**Fig 4. Box-Plot of the ED patients by meteorological features.**

## Building the ANN

The ANN-ELM built using a conventional back-propagation algorithm, with a single hidden layer. The ANN algorithm built with "nnfor" package in R4.3.2. The choice of ANN_ELM structure is concerned with determining input features and hidden nodes. A total of 17 calendar and meteorological features in Table 1, taken into account in the ANN. The number of hidden nodes was determined to be 40 by the package algorithm to optimize the algorithm based on the minimum RMSE loss function.

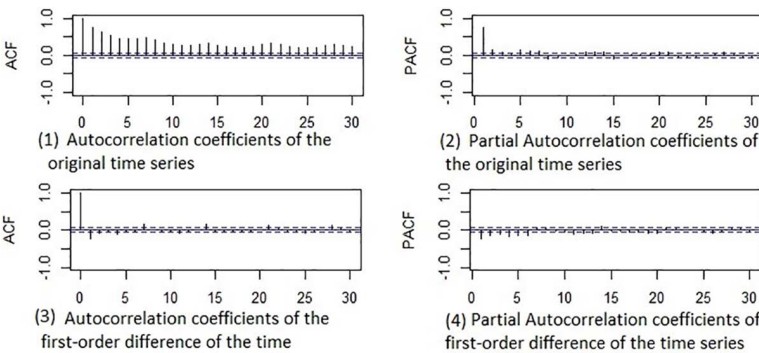

**Fig 5. Sample ACF and PACF of daily number of ED patients.**

**Table 2. Results of ARIMAX.**

| Feature | Unstandardized | | Standardized | | P-value |
|---|---|---|---|---|---|
| | Coefficient | S.E of Coefficient | Coefficient | S.E of Coefficient | |
| AR1 | 0.862 | 0.031 | 0.863 | 0.032 | 0.000 |
| MA1 | −1.451 | 0.049 | −1.466 | 0.05 | 0.000 |
| MA2 | 0.455 | 0.047 | 0.471 | 0.048 | 0.000 |
| X2 | −35.413 | 10.391 | −0.532 | 0.162 | 0.001 |
| X5 | 19.396 | 2.823 | 0.308 | 0.044 | 0.000 |
| X6 | 25.573 | 3.484 | 0.409 | 0.054 | 0.000 |
| X7 | 24.039 | 3.448 | 0.377 | 0.054 | 0.000 |
| X8 | 22.825 | 3.761 | 0.325 | 0.058 | 0.000 |
| X11 | 3.368 | 0.395 | 0.055 | 0.006 | 0.000 |
| X13 | 0.934 | 0.556 | 0.015 | 0.009 | 0.082 |

Feature importance analysis reveals substantial divergence between the ANN and ARIMAX models. In contrast to ARIMAX, which only identified average temperature (X11) as a significant meteorological predictor (Table 2), the ANN model places greater weight on meteorological factors than calendar factors. Notably, average humidity (X16) emerged as the most critical feature for the ANN (Fig. 6), despite its lack of significance in the ARIMAX framework.

## Building the first hybrid algorithm

Hybrid1 model was also an ANN built using the following input features; the fitted values of the ANN, the fitted values of the ARIMAX, the fitted values of the ANN on the residual of the ARIMAX, and the nine input features that excluded from the ARIMAX. To determine the number of hidden nodes, an automated approach according to the root mean squared error (RMSE) criterion performed. Finally, 40 hidden nodes selected for a hidden layer. As shown in Table 3, the forecast accuracy of this algorithm is better than ARIMA and ANN model for short and intermediate horizons.

## Building the second hybrid algorithm

Hybrid2 algorithm used of two separate ARIMAX and ANN algorithms. To obtain the forecast of hybrid2 algorithm the mean of forecasting values from ANN and ARIMAX algorithm calculated.

Fig 7 displays the forecasting plots for emergency department patients using ARIMA, ANN, and two hybrid ARIMA-ANN algorithms alongside actual data. Fig 7(1) shows that ARIMAX forecasts are generally tend to be underestimated, while

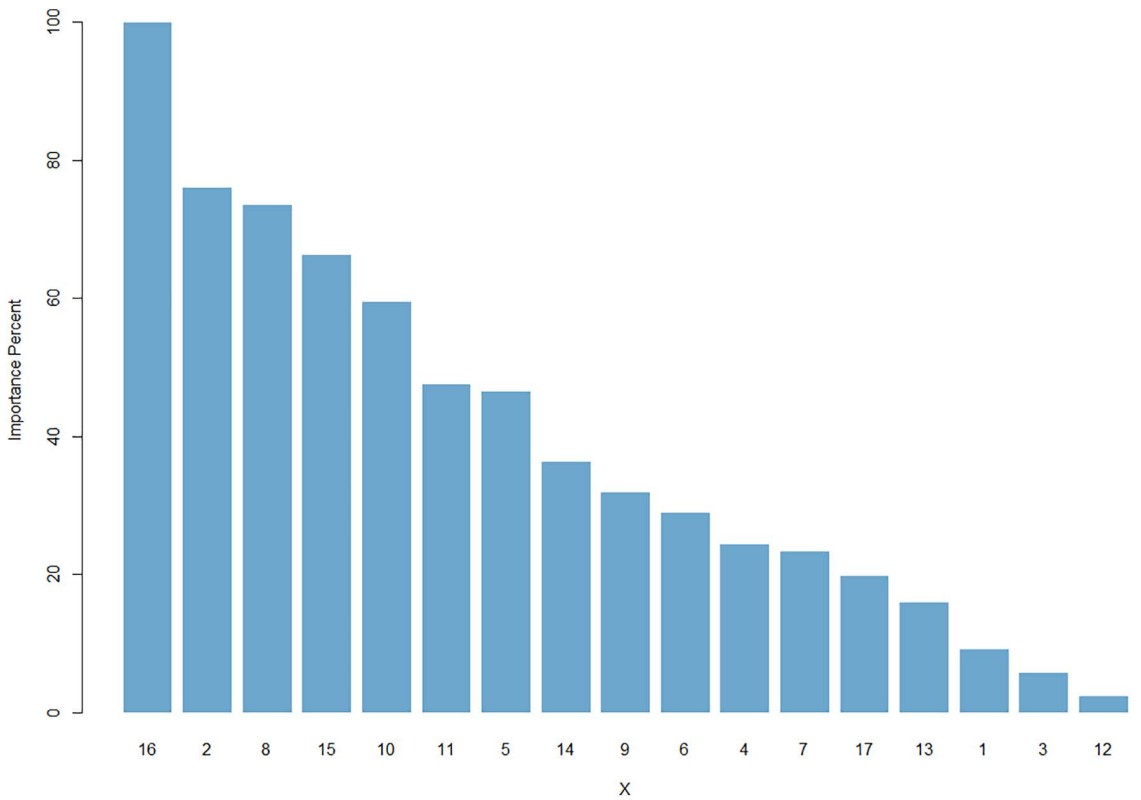

**Fig 6. Feature importance in ANN.**

Fig 7(2) reveals that ANN forecasts tend to be overestimated, with the forecasts from both hybrid algorithms Figs 7(3) and 7(4) falling between ARIMAX and ANN forecasts.

## Building the LSTM

The Long Short-Term Memory (LSTM) network was implemented using Python with the TensorFlow/Keras library, utilizing the "LSTM", "Dense", and "Dropout" layers for its structure and regularization. The input configuration for the recurrent layer was defined with a shape of (30,17), representing a lookback window of 30 time steps across 17 input features. For optimization, the model was compiled using the Adam optimizer and the Mean Squared Error (MSE) loss function, a standard choice for regression problems. The network was subsequently trained for 50 epochs using a batch size of 30. Upon completion of training, the established model was employed to generate scaled predictions corresponding to the final 84 time steps isolated within the designated test set. The forecasting accuracy indices of this algorithm show in table 3.

## Building the GLM

The GLM built using Negative-Binomial distribution for ED arrivals and identity link function. The GLM algorithm built with "MASS" package in R4.3.2. The choice of GLM structure is concerned with determining appropriate distribution function, link function, and input features. Among of 17 calendar and meteorological features in Table 1, forward feature selection according to MICs, taken into account in the GLM. The forecasting accuracy indices of this model show in Table 3.

**Table 3. Comparing the accuracy of forecasting with ARIMA, ANN, two hybrid algorithms, LSTM and GLM.**

| Forecast Horizons | Model | Forecast Accuracy Indices | | | | | | | Diebold-Mariano Test | |
|---|---|---|---|---|---|---|---|---|---|---|
| | | R² | ME | RMSE | MAE | MPE | MAPE | SMAPE | DM | p |
| Short: 1-28 days | ARIMA | 0.08 | −21.79 | 37.75 | 31.98 | −4.2 | 6.1 | 5.91 | 3.75 | 0.001 |
| | ANN | 0.01 | 10.84 | 39.28 | 31.41 | 1.76 | 5.53 | 5.67 | 1.50 | 0.156 |
| | Hybrid1 | 0.04 | 0.89 | 33.61 | 27.55 | 0.05 | 4.98 | 5.03 | 0.07 | 0.943 |
| | Hybrid2 | 0.03 | −5.48 | 34.16 | 29.68 | −1.11 | 5.45 | 5.43 | 0.92 | 0.367 |
| | LSTM | 0.01 | −43.24 | 58.14 | 49.68 | −8.79 | 9.99 | 9.38 | 5.76 | 0.001 |
| | GLM | 0.12 | 13.83 | 32.40 | 27.68 | 2.27 | 5.01 | 5.28 | 2.56 | 0.016 |
| Intermediate: 29–56 days | ARIMA | 0.63 | −28.44 | 45.78 | 34.26 | −5.27 | 6.43 | 6.12 | 4.11 | 0.001 |
| | ANN | 0.27 | 35.05 | 56.89 | 48.98 | 5.96 | 8.34 | 8.78 | 3.91 | 0.001 |
| | Hybrid1 | 0.53 | 9.86 | 41.73 | 35.04 | 1.86 | 6.22 | 6.33 | 0.99 | 0.331 |
| | Hybrid2 | 0.60 | 3.31 | 38.53 | 31.4 | 0.72 | 5.61 | 5.65 | 0.16 | 0.868 |
| | LSTM | 0.16 | −49.18 | 68.62 | 53.51 | −9.75 | 10.61 | 9.8 | 6.80 | 0.001 |
| | GLM | 0.44 | 15.96 | 42.55 | 33.92 | 2.36 | 5.89 | 5.13 | 1.40 | 0.173 |
| Long: 57-84 days | ARIMA | 0.37 | 1.09 | 26.13 | 21.13 | 0.15 | 3.94 | 3.94 | 0.15 | 0.884 |
| | ANN | 0.33 | 20.35 | 33.41 | 28.21 | 3.66 | 5.11 | 5.27 | 3.94 | 0.001 |
| | Hybrid1 | 0.23 | 31.05 | 42.08 | 35.83 | 5.5 | 6.36 | 6.64 | 5.67 | 0.001 |
| | Hybrid2 | 0.36 | 10.72 | 27.94 | 23.87 | 1.95 | 4.39 | 4.47 | 2.06 | 0.049 |
| | LSTM | 0.19 | −16.3 | 34.09 | 29.6 | −3.25 | 5.75 | 5.62 | 2.88 | 0.008 |
| | GLM | 0.37 | −12.83 | 28.39 | 24.34 | −2.63 | 4.66 | 6.11 | 2.52 | 0.018 |
| Overall: 1-84 days | ARIMA | 0.32 | −16.38 | 37.43 | 29.12 | −3.1 | 5.49 | 5.32 | 4.53 | 0.001 |
| | ANN | 0.44 | 22.08 | 44.33 | 36.2 | 3.79 | 6.33 | 6.57 | 5.11 | 0.001 |
| | Hybrid1 | 0.19 | 13.93 | 39.33 | 32.81 | 2.47 | 5.85 | 6.00 | 3.18 | 0.002 |
| | Hybrid2 | 0.32 | 2.85 | 33.82 | 28.32 | 0.52 | 5.15 | 5.18 | 0.49 | 0.629 |
| | LSTM | 0.06 | −36.24 | 55.53 | 44.26 | −7.26 | 8.78 | 8.27 | 7.77 | 0.001 |
| | GLM | 0.29 | 5.65 | 34.96 | 28.65 | 0.66 | 5.18 | 4.62 | 1.71 | 0.091 |

## Comparing the short, intermediate and long term forecast accuracies of algorithms

Table 3 presents the accuracy of forecasting of ARIMAX, ANN, two hybrid algorithms, LSTM, and GLM using seven accuracy indices across short, intermediate and long horizons. The analysis reveals the following key findings:

1. Short horizon accuracy: The Hybrid1 algorithm achieved a SMAPE of 5.03, MAPE = 4.98 and an AME of 27.55, which are the best among the indices. Consequently, the Hybrid2 algorithm outperformed both the ARIMAX and ANN algorithms, demonstrating its effectiveness in capturing short-term patient arrival trends. Although the DM test does not indicate a significant difference between the observed and forecasted values produced by the ANN, Hybrid1, and Hybrid2, all accuracy indices—except $R^2$—show that the Hybrid algorithms outperform the GLM. These results suggest that integrating features from both algorithms enhances forecasting precision in the short horizon.

2. Intermediate horizon accuracy: The high volatility in ED patient arrivals was in this horizon and the forecasts of ANN and ARIMAX were completely different. The ME of ARIMAX was −28.44 (vigorous underestimation) and the ME of ANN was 35.05 (vigorous overestimation). In this horizon Hybrid2 algorithm with ME = 3.31, SMAPE = 5.65, and MAPE = 5.61 consistently outperformed all other algorithms. Although the DM test does not indicate a significant difference between the observed and forecasted values produced by the Hybrid1, Hybrid2, and GLM algorithms, all accuracy indices show that the Hybrid algorithms outperform the ANN, ARIMAX, LSTM, and GLM.

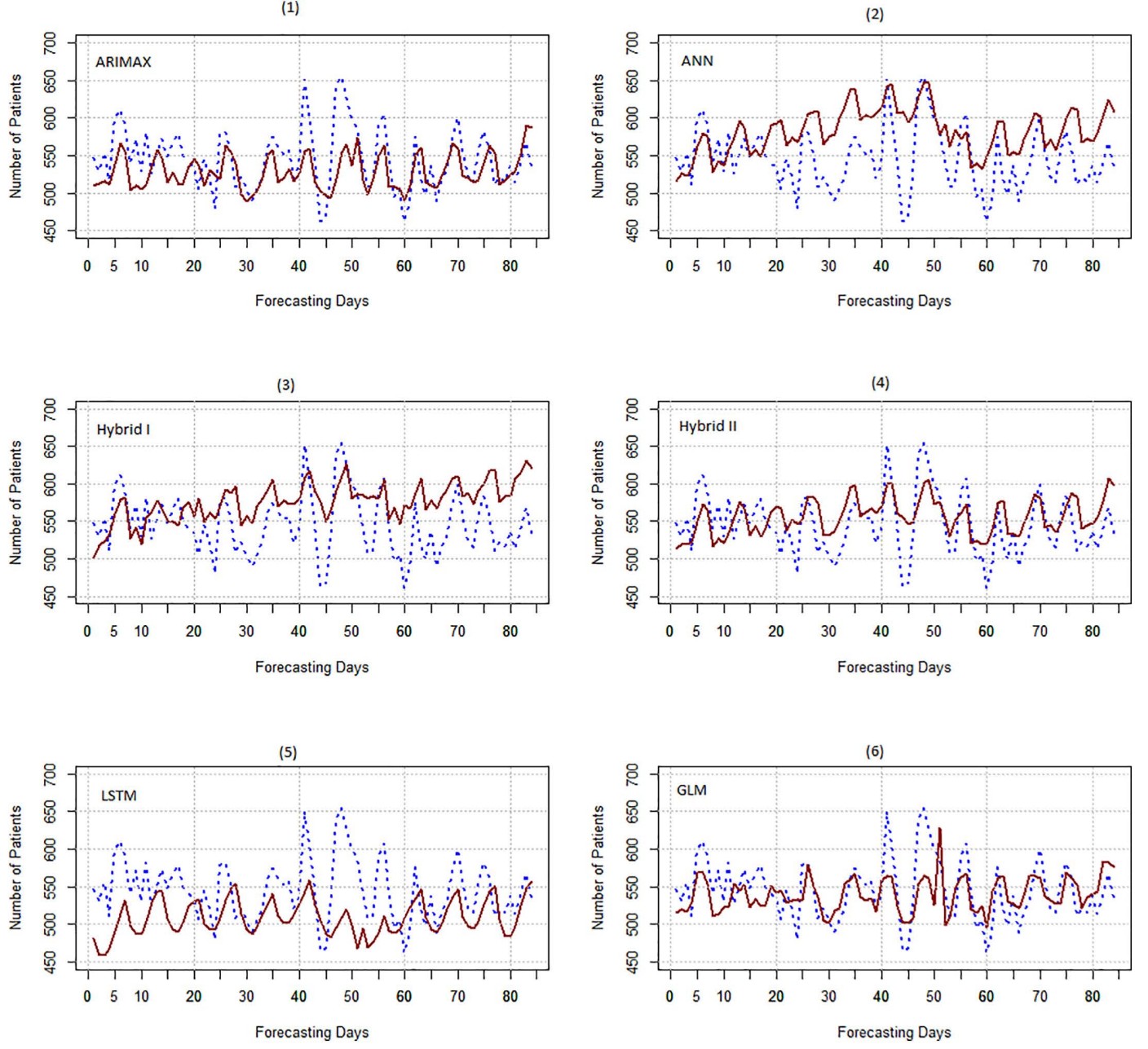

**Fig 7. Comparing the forecasts of ARIMAX, ANN, two hybrid algorithms, LSTM, and GLM with actual data.**

3. Long horizon accuracy: The low volatility in ED patient arrivals was in this horizon, the ARIMAX algorithms demonstrated superior accuracy with ME = 1.09 and MAPE = SMAPE = 3.94 compared to other algorithms. At this forecasting horizon, although the accuracy indices of Hybrid2 are satisfactory, the DM test indicates a significant difference between the observed and forecasted values for all algorithms except ARIMAX.

4. Overall accuracy: Over the multiple horizons of 0–12 weeks (84 days), the Hybrid2 algorithm demonstrated superior forecasting accuracy with ME = 2.82, SMAPE = 5.18 and MAPE = 5.15 compared to ARIMAX, ANN, Hybrid1, LSTM, and

GLM. This indicates that Hybrid2 effectively captures overall patterns in patient arrivals. Across the overall forecasting horizon, although the DM test indicates no significant difference between the observed and forecasted values for the GLM model, the accuracy indices of the Hybrid2 algorithm demonstrate its superior performance. This consistent accuracy underscores the robustness of the Hybrid2 algorithm across varying forecasting horizons.

## Discussion

The primary contribution of the present study lies in providing an accurate, context-driven empirical forecast of emergency department (ED) arrivals by building upon established ARIMA–ANN frameworks. It offers new evidence on the applicability and performance of these hybrid models in a highly volatile healthcare environment. Three aspects distinguish this work from previous studies; First, it delivers a comprehensive evaluation of hybrid time-series models in the context of Mashhad, Iran, where patient arrivals are strongly influenced by religious holidays, tourism, and diverse seasonal patterns. Second, use of the Maximal Information Coefficient (MIC) as a data-driven approach for selecting both calendar and meteorological predictors, ensuring that only meaningful exogenous variables are incorporated into the ARMAX model. Third, by analyzing forecast performance across short-, intermediate-, and long-term horizons, the study demonstrates how such combinations can improve the stability and operational usefulness of forecasts for emergency department (ED) management. Rather than proposing fundamentally new algorithms, this research advances practical understanding by adapting and validating hybrid forecasting strategies for a regional ED setting with high variability, thereby offering evidence-based guidance for data-informed healthcare planning and decision-making.

Accurate forecasts of future patients' arrivals in emergency departments are necessary for health decision processes [18]. The present study specifically addresses the context of emergency departments in Iran, considering local factors and characteristics that may influence patient arrivals. Mashhad is the second largest city in Iran, many people travel to this city because of its religious and tourist attractions. For this reason, the number of referees to the emergency department suddenly increases during holidays and religious occasions, so the volatility in their time series data is extremely high. This city has really four seasons and the weather changes accordingly. The weather in spring is very nice for travel destinations but in summer, it has a hot month. At the half of the last month of summer, autumn comes with a cool breeze and then a cold and dry winter began. Therefore, this study emphasizes the inclusion of both calendar and meteorological features as significant predictors for patient arrivals [8–13,19]. The advanced statistical techniques for data preprocessing, including the use of MIC and modified FDR control procedures to adjust p-values for multiple comparisons is the highlights of this study. This nonparametric method allows for a comprehensive exploration of the relationships between input features and patient arrivals, accommodating nonlinear associations that traditional correlation methods might miss [10,20]. The role of MIC-based feature selection in this study should be interpreted with caution. MIC was applied exclusively to ARIMAX and GLM models to enhance model stability and interpretability by limiting the inclusion of highly dependent or weakly informative exogenous variables. In contrast, ANN and LSTM models were trained using the full set of calendar and meteorological features, leveraging their ability to handle complex and high-dimensional inputs. Consequently, the performance gains observed in hybrid models—particularly Hybrid2—cannot be attributed solely to MIC-based feature selection, but rather to the integration of complementary modeling structures and the statistical stabilization achieved through combining linear and nonlinear forecasts.

The results of this study offer strong evidence that standalone ARIMAX and ANN models lack sufficient accuracy, highlighting the need for hybrid forecasting algorithms to predict daily patient arrivals in Emergency Departments (ED). Combining Auto-Regressive Integrated Moving Average (ARIMA/ARIMAX) with Artificial Neural Networks (ANN) not only improves predictive accuracy but also tackles the complexities of patient arrival patterns affected by various external factors. These conclusions align with findings from earlier studies [14–16]. Compared to the hybrid algorithm proposed by Zhang et al. (2003), this algorithm offers greater flexibility in utilizing the residuals of ARIMAX. Since ARIMAX tends to underestimate forecasts, its residuals may exhibit potential bias. Consequently, incorporating forecasts based on the ANN

algorithm into the ARIMAX residuals could not enhance the accuracy of the ARIMAX forecast for the current time series [1]. In our study, the kurtosis of the ARIMAX and ANN residuals, influenced by the exogenous input features, was not significantly greater than 3. Therefore, the hybrid approach proposed by Babu (2014), which suggests using an MA filter for high-low volatility decomposition in time series [2], is not suitable.

The performance of Hybrid2 Algorithm, which combines the strengths of both ARIMA and ANN, illustrates the advantages of utilizing hybrid approaches in time series forecasting. Traditional ARIMA algorithms are adept at capturing linear relationships within time series data, while ANN excels in capturing nonlinear patterns [1–3]. The importance of the input features for ARIMAX and ANN, illustrated in Figure..., reveals a complex relationship between these features and the raw time series data. Some of these relationships identified by ARIMAX, while ANN captured others. By combining these algorithms, the hybrid models can effectively address the intricate nature of patient arrivals, which frequently display both linear trends and nonlinear fluctuations due to seasonal variations, holidays, and other external factors [14–16].

The significant impact of calendar and meteorological features on patient arrivals highlights the necessity of incorporating contextual factors into forecasting algorithms. For instance, variations in weather conditions, such as extreme temperatures or precipitation, can influence the frequency of ED visits, as certain health issues may have exacerbated by specific weather patterns. Similarly, calendar events, including weekends, holidays, and local festivities, tend to affect patient behavior and demand for emergency services. This study underscores the importance of local context in predictive algorithms, suggesting that hospitals should tailor their algorithms to reflect the unique characteristics of their patient populations and regional influences, which supported by previous studies [14,16–20].

The implications of this research extend beyond theoretical advancements. The results offer practical solutions for healthcare administrators. Accurate forecasting of patient arrivals is crucial for optimizing resource allocation, staffing, and operational efficiency within emergency departments. By utilizing the proposed hybrid algorithms, hospitals can better anticipate fluctuations in patient volume, thereby enhancing their ability to manage peak times and reduce wait times. This proactive approach can lead to improved patient satisfaction and outcomes; as timely care is essential in emergency settings [14–22].

Despite the promising results, this study acknowledges certain limitations. The reliance on historical data from a specific timeframe may not capture sudden changes in patient behavior due to unforeseen circumstances, such as pandemics or public health crises. Therefore, continuous algorithm updating and recalibration will be necessary to maintain accuracy over time. Additionally, the complexity of the algorithms may pose challenges in terms of implementation for smaller healthcare facilities with limited resources and expertise in advanced analytics.

While this study provides a solid foundation for the application of hybrid forecasting algorithms in healthcare, several avenues for future research remain. Expanding the dataset to include additional features—such as socioeconomic indicators, public health trends, and historical patient data—could further enhance algorithm accuracy. Additionally, exploring the integration of machine learning techniques such as ensemble methods including ARIMA and other statistical methods yield to more robust predictive capabilities.

In summary, the results of this study provide strong evidence for the superiority of hybrid ARIMA-ANN algorithms in forecasting daily patient arrivals in Emergency Departments. The significant improvements in accuracy and reliability highlight the potential of these methodologies to transform healthcare analytics and support informed decision-making in emergency care settings.

## Conclusion

This study demonstrates that both hybrid algorithms outperform the individual ARIMAX and ANN algorithms across all accuracy metrics. Notably, Hybrid2 achieves the highest overall performance, displaying the effectiveness of combining the strengths of ARIMAX and ANN to tackle the complexities of patient arrival patterns. The reduction in error metrics associated with the hybrid algorithms highlights their ability to enhance forecasting accuracy. By integrating both ARIMA and ANN methodologies, these algorithms are better position to capture the nuances of patient arrivals, leading to

forecasts that are more reliable. This analysis suggests that adopting hybrid approaches can significantly improve forecasting efforts in healthcare settings, especially in emergency departments where accurate predictions are essential for resource allocation and patient care.

The analysis also revealed that both calendar and meteorological features are crucial in influencing patient arrivals. By incorporating these features into the forecasting algorithms, the hybrid models effectively captured the complexities and nonlinearities present in patient arrival data. This integration not only enhanced the predictive capabilities of the algorithms but also provided valuable insights into the factors affecting patient flow in the emergency department.

The implications of this research are significant for healthcare decision-makers. Accurate forecasting of patient arrivals is vital for optimizing staffing, managing medical supplies, and reducing wait times in emergency care settings. The proposed hybrid algorithms can serve as valuable tools for hospital administrators and policymakers, enabling informed decisions based on reliable forecasts. Utilizing short, intermediate, and long-term forecasts is crucial for optimizing staff rotations and schedules. By leveraging these diverse time horizons, organizations can ensure that they have the right number of staff available to meet varying demands throughout the day, week, or month. Short-term forecasts allow for immediate adjustments, while intermediate and long-term forecasts help in strategic planning and resource allocation.

Additionally, it is essential to account for influential factors that can change abruptly, such as weather conditions, public events, or unexpected surges in patient arrivals. These factors can significantly influence staffing needs and operational efficiency. Therefore, updating forecasts regularly in response to these changes is vital. Implementing a dynamic forecasting system that incorporates real-time data can enhance responsiveness and ensure that staffing levels remain aligned with actual demand.

Moreover, engaging with staff to gather insights on their availability and preferences can further refine scheduling practices. This collaborative approach not only improves staff satisfaction but also enhances overall service delivery. By integrating predictive analytics with flexible scheduling strategies, organizations can better navigate the complexities of staffing in a rapidly changing environment, ultimately leading to improved operational effectiveness and enhanced patient care. Furthermore, the study emphasizes the importance of local calibration in predictive algorithms. Given the variability in patient arrivals influenced by cultural, seasonal, and meteorological factors, algorithms must be tailor to reflect the specific characteristics of each healthcare facility. Future research should investigate the application of these hybrid algorithms in various geographical contexts and expand the dataset to include additional input features, such as socioeconomic factors and public health events.

## Supporting information

**S1 File. R and Python Codes.**
(ZIP)

**S2 File. Integrated Data.**
(CSV)

## Acknowledgments

This paper is an extension of MSc thesis of AAM under the supervision of AS at the Department of Epidemiology and Biostatistics, School of Health, Mashhad University of Medical Sciences.

## Author contributions

**Data curation:** Ali AbbaszadehMozaffari, Zahra Ebnehoseini.

**Formal analysis:** Ali AbbaszadehMozaffari.

**Methodology:** Hamed Tabesh, Azadeh Saki.

**Supervision:** Azadeh Saki.

**Writing – original draft:** Azadeh Saki.

**Writing – review & editing:** Hamed Tabesh, Azadeh Saki.

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
