## [Decision Letter · Decision Letter 0]

23 Sep 2025

PONE-D-25-37594Comparison of ARIMAX, Artificial Neural Networks and Hybrid ARIMAX-ANN Algorithms for Forecasting the Daily Number of Patients Arrivals in Emergency DepartmentPLOS ONE

Dear Dr. Saki,

Thank you for submitting your manuscript to PLOS ONE. After careful consideration, we feel that it has merit but does not fully meet PLOS ONE’s publication criteria as it currently stands. Therefore, we invite you to submit a revised version of the manuscript that addresses the points raised during the review process.

If applicable, we recommend that you deposit your laboratory protocols in protocols.io to enhance the reproducibility of your results. Protocols.io assigns your protocol its own identifier (DOI) so that it can be cited independently in the future. For instructions see: https://journals.plos.org/plosone/s/submission-guidelines#loc-laboratory-protocols. Additionally, PLOS ONE offers an option for publishing peer-reviewed Lab Protocol articles, which describe protocols hosted on protocols.io. Read more information on sharing protocols at . Additionally, PLOS ONE offers an option for publishing peer-reviewed Lab Protocol articles, which describe protocols hosted on protocols.io. Read more information on sharing protocols at https://plos.org/protocols?utm_medium=editorial-email&utm_source=authorletters&utm_campaign=protocols..

We look forward to receiving your revised manuscript.

Kind regards,

Youngsang Cho

Academic Editor

PLOS ONE

Additional Editor Comments:

Reviewer #1:

This paper appears to be an attempt to propose an interesting model in the field of time series prediction, but it is not suitable for publication in its current state due to insufficient presentation of the problem definition, inadequate comparative experiments, limitations in performance evaluation, and lack of reproducibility.

In particular, it is essential to conduct a fair comparison with the latest time series models, clarify the hyperparameter optimisation process, introduce various performance metrics and statistical tests, and ensure the interpretability of variables. Additionally, the paper must present its applicability and research contribution in a more persuasive manner.

Therefore, this paper requires a major revision, and it is judged that it will only have publication value if the aforementioned issues are thoroughly addressed.

1. This paper aims to improve performance in specific prediction problems (time series-based), but it is unclear how it specifically fills the gap compared to existing research. It must be clarified whether it simply applies a new model or modifies an existing model, or whether it provides new insights in actual industrial/social applications. The motivation for ‘why this research is necessary’ is lacking.

2. The description of the data set's source, collection process, and preprocessing methods (missing value handling, outlier removal, normalisation, etc.) is insufficient. In particular, in time series prediction research, the periodicity, seasonality, and trend characteristics of the data are important, and the results of exploratory analysis (ACF, PACF, time series decomposition, etc.) should be presented. Otherwise, it is difficult to judge the validity of the modelling.

3. The paper mentions hyperparameter settings during model training, but the rationale is insufficient. It should clearly state whether the default values were used or whether methods such as grid search or Bayesian optimisation were applied. The performance of time series prediction models is sensitive to parameters such as learning rate, window size, and hidden dimension, so failure to address these systematically weakens the credibility of the research.

4. Only a few evaluation metrics such as RMSE or MAE were used, which are insufficient to adequately address model characteristics and problem definitions. It is necessary to present various metrics such as MAPE, SMAPE, and R² to verify prediction bias. In addition, statistical significance tests (e.g., Diebold-Mariano test) should be used to prove that the performance differences between models are meaningful.

5. The paper focuses solely on improving model performance, but does not analyse which variables (features) contributed significantly to the prediction. In particular, time series prediction research needs to strengthen its explanatory power through variable importance (feature importance) and attention weight visualisation when considering practical application. Simply stating that ‘the accuracy is high’ limits the academic and practical contributions.

6. The reproducibility is low because the experimental code, parameter details, and hardware/software environment are not specified. In particular, ensuring reproducibility is important in the latest time series model research, but this paper overlooks this point. At the very least, pseudocode, data split method (train/valid/test ratio), and random seed settings should be provided.

7. In the literature review, recent studies on deep learning models for time series prediction published in the last two to three years were not sufficiently cited. In particular, studies on Transformer-based prediction were omitted, which is a major flaw in clearly establishing the position of this study in academia.

8. Lack of discussion on how the model presented in the paper can be applied in actual industries, policies, or services. It is difficult to judge the application value based solely on the result that ‘prediction accuracy has improved.’

Reviewer #2:

This manuscript addresses a highly relevant and practice-oriented issue: forecasting daily patient arrivals in emergency departments (EDs) by combining time series and machine learning techniques. The integration of calendar and meteorological variables with both ARIMAX and ANN, along with the development of two hybrid algorithms, is a valuable contribution to healthcare operations research. The comparative evaluation across multiple horizons (short, intermediate, and long term) further enhances the practical applicability of the findings. That said, several conceptual and methodological issues should be addressed to maximize the clarity and impact of the study. The manuscript would benefit from a more structured framing of the literature review, stronger justification for benchmark model selection, clearer reporting of ANN architecture and hyperparameter optimization, and deeper interpretation of horizon-specific performance differences. Additionally, highlighting the unique contributions of this study—beyond confirming the established role of calendar and meteorological factors—would strengthen its significance.

- The manuscript cites a large number of related studies in the introduction; however, these references are presented in a rather descriptive manner, without offering a coherent narrative. Summarizing the common implications and insights of prior research, and explicitly linking their limitations to the distinctive contributions of this study, would make the introduction more structured and persuasive.

- The study proposes two hybrid models and compares them with ARIMA and ANN. However, alternative hybrid models proposed in previous studies, such as the MA-filter based hybrid ARIMA–ANN algorithm (Babu et al., 2014) and the fuzzy time series algorithm (Jilani et al., 2019), were not included as benchmarks. The rationale for excluding these models should be clarified in more detail.

- The manuscript mentions that 40 hidden nodes were used in the ANN model, but it is unclear whether the number of hidden layers was fixed at one. If a single hidden layer was adopted, was this the result of hyperparameter optimization or the authors’ discretion? Furthermore, discussing how ANN performance changes with different configurations of hidden layers and nodes would strengthen the robustness of the modeling approach.

- Figure 6 shows that the variable importance differs across ARIMAX and ANN, even when the same variables are used. An explanation of whether this difference stems from the contrast between linear and nonlinear approaches, or from other factors, would enhance the interpretability of the results.

- Typically, ANN analyses utilize a broad set of input variables to leverage its predictive capacity. In contrast, this study employed the Maximal Information Coefficient (MIC) approach for variable selection. Providing comparative evidence of model performance with and without the MIC approach would make the methodological contribution clearer.

- It remains unclear how the dataset was divided into training, validation, and testing sets, and whether repeated experiments were conducted to ensure robustness. Since data splitting strategies and hyperparameter tuning can significantly affect model performance, Table 2 alone may not provide sufficient evidence to conclude that one model is superior to another.

- The results show that the best-performing model differs by horizon: Hybrid1 performs best in the short horizon, whereas Hybrid2 performs better in the intermediate horizon. The reasons for these differences should be discussed in more depth. Additionally, from a practitioner’s perspective, guidance is needed on which model to use for short-, medium-, and long-term scheduling. If different models are to be applied depending on the time horizon, recommendations on how to integrate their outputs would be valuable.

- The finding that calendar and meteorological variables significantly influence patient arrivals is consistent with prior literature and may even be considered self-evident. The conclusion would be strengthened by emphasizing what is newly discovered in this study or highlighting its unique contributions beyond confirming established results.

Reviewers' comments:

Reviewer's Responses to Questions

**Comments to the Author**

1. Is the manuscript technically sound, and do the data support the conclusions?

Reviewer #1: Yes

Reviewer #2: Yes

2. Has the statistical analysis been performed appropriately and rigorously? 

Reviewer #1: Yes

Reviewer #2: No

3. Have the authors made all data underlying the findings in their manuscript fully available?

Reviewer #1: No

Reviewer #2: No

4. Is the manuscript presented in an intelligible fashion and written in standard English?

Reviewer #1: Yes

Reviewer #2: Yes

5. Review Comments to the Author

Reviewer #1: This paper appears to be an attempt to propose an interesting model in the field of time series prediction, but it is not suitable for publication in its current state due to insufficient presentation of the problem definition, inadequate comparative experiments, limitations in performance evaluation, and lack of reproducibility.

In particular, it is essential to conduct a fair comparison with the latest time series models, clarify the hyperparameter optimisation process, introduce various performance metrics and statistical tests, and ensure the interpretability of variables. Additionally, the paper must present its applicability and research contribution in a more persuasive manner.

Therefore, this paper requires a major revision, and it is judged that it will only have publication value if the aforementioned issues are thoroughly addressed.

1. This paper aims to improve performance in specific prediction problems (time series-based), but it is unclear how it specifically fills the gap compared to existing research. It must be clarified whether it simply applies a new model or modifies an existing model, or whether it provides new insights in actual industrial/social applications. The motivation for ‘why this research is necessary’ is lacking.

2. The description of the data set's source, collection process, and preprocessing methods (missing value handling, outlier removal, normalisation, etc.) is insufficient. In particular, in time series prediction research, the periodicity, seasonality, and trend characteristics of the data are important, and the results of exploratory analysis (ACF, PACF, time series decomposition, etc.) should be presented. Otherwise, it is difficult to judge the validity of the modelling.

3. The paper mentions hyperparameter settings during model training, but the rationale is insufficient. It should clearly state whether the default values were used or whether methods such as grid search or Bayesian optimisation were applied. The performance of time series prediction models is sensitive to parameters such as learning rate, window size, and hidden dimension, so failure to address these systematically weakens the credibility of the research.

4. Only a few evaluation metrics such as RMSE or MAE were used, which are insufficient to adequately address model characteristics and problem definitions. It is necessary to present various metrics such as MAPE, SMAPE, and R² to verify prediction bias. In addition, statistical significance tests (e.g., Diebold-Mariano test) should be used to prove that the performance differences between models are meaningful.

5. The paper focuses solely on improving model performance, but does not analyse which variables (features) contributed significantly to the prediction. In particular, time series prediction research needs to strengthen its explanatory power through variable importance (feature importance) and attention weight visualisation when considering practical application. Simply stating that ‘the accuracy is high’ limits the academic and practical contributions.

6. The reproducibility is low because the experimental code, parameter details, and hardware/software environment are not specified. In particular, ensuring reproducibility is important in the latest time series model research, but this paper overlooks this point. At the very least, pseudocode, data split method (train/valid/test ratio), and random seed settings should be provided.

7. In the literature review, recent studies on deep learning models for time series prediction published in the last two to three years were not sufficiently cited. In particular, studies on Transformer-based prediction were omitted, which is a major flaw in clearly establishing the position of this study in academia.

8. Lack of discussion on how the model presented in the paper can be applied in actual industries, policies, or services. It is difficult to judge the application value based solely on the result that ‘prediction accuracy has improved.’

Reviewer #2: This manuscript addresses a highly relevant and practice-oriented issue: forecasting daily patient arrivals in emergency departments (EDs) by combining time series and machine learning techniques. The integration of calendar and meteorological variables with both ARIMAX and ANN, along with the development of two hybrid algorithms, is a valuable contribution to healthcare operations research. The comparative evaluation across multiple horizons (short, intermediate, and long term) further enhances the practical applicability of the findings. That said, several conceptual and methodological issues should be addressed to maximize the clarity and impact of the study. The manuscript would benefit from a more structured framing of the literature review, stronger justification for benchmark model selection, clearer reporting of ANN architecture and hyperparameter optimization, and deeper interpretation of horizon-specific performance differences. Additionally, highlighting the unique contributions of this study—beyond confirming the established role of calendar and meteorological factors—would strengthen its significance.

- The manuscript cites a large number of related studies in the introduction; however, these references are presented in a rather descriptive manner, without offering a coherent narrative. Summarizing the common implications and insights of prior research, and explicitly linking their limitations to the distinctive contributions of this study, would make the introduction more structured and persuasive.

- The study proposes two hybrid models and compares them with ARIMA and ANN. However, alternative hybrid models proposed in previous studies, such as the MA-filter based hybrid ARIMA–ANN algorithm (Babu et al., 2014) and the fuzzy time series algorithm (Jilani et al., 2019), were not included as benchmarks. The rationale for excluding these models should be clarified in more detail.

- The manuscript mentions that 40 hidden nodes were used in the ANN model, but it is unclear whether the number of hidden layers was fixed at one. If a single hidden layer was adopted, was this the result of hyperparameter optimization or the authors’ discretion? Furthermore, discussing how ANN performance changes with different configurations of hidden layers and nodes would strengthen the robustness of the modeling approach.

- Figure 6 shows that the variable importance differs across ARIMAX and ANN, even when the same variables are used. An explanation of whether this difference stems from the contrast between linear and nonlinear approaches, or from other factors, would enhance the interpretability of the results.

- Typically, ANN analyses utilize a broad set of input variables to leverage its predictive capacity. In contrast, this study employed the Maximal Information Coefficient (MIC) approach for variable selection. Providing comparative evidence of model performance with and without the MIC approach would make the methodological contribution clearer.

- It remains unclear how the dataset was divided into training, validation, and testing sets, and whether repeated experiments were conducted to ensure robustness. Since data splitting strategies and hyperparameter tuning can significantly affect model performance, Table 2 alone may not provide sufficient evidence to conclude that one model is superior to another.

- The results show that the best-performing model differs by horizon: Hybrid1 performs best in the short horizon, whereas Hybrid2 performs better in the intermediate horizon. The reasons for these differences should be discussed in more depth. Additionally, from a practitioner’s perspective, guidance is needed on which model to use for short-, medium-, and long-term scheduling. If different models are to be applied depending on the time horizon, recommendations on how to integrate their outputs would be valuable.

- The finding that calendar and meteorological variables significantly influence patient arrivals is consistent with prior literature and may even be considered self-evident. The conclusion would be strengthened by emphasizing what is newly discovered in this study or highlighting its unique contributions beyond confirming established results.

6. PLOS authors have the option to publish the peer review history of their article (what does this mean?). If published, this will include your full peer review and any attached files.). If published, this will include your full peer review and any attached files.

.

Reviewer #1: No

Reviewer #2: No

While revising your submission, please upload your figure files to the Preflight Analysis and Conversion Engine (PACE) digital diagnostic tool, https://pacev2.apexcovantage.com/. PACE helps ensure that figures meet PLOS requirements. To use PACE, you must first register as a user. Registration is free. Then, login and navigate to the UPLOAD tab, where you will find detailed instructions on how to use the tool. If you encounter any issues or have any questions when using PACE, please email PLOS at . PACE helps ensure that figures meet PLOS requirements. To use PACE, you must first register as a user. Registration is free. Then, login and navigate to the UPLOAD tab, where you will find detailed instructions on how to use the tool. If you encounter any issues or have any questions when using PACE, please email PLOS at figures@plos.org. Please note that Supporting Information files do not need this step.. Please note that Supporting Information files do not need this step.

---

## [Author Response · Author response to Decision Letter 1]

11 Dec 2025

Reviewer #1:

This paper appears to be an attempt to propose an interesting model in the field of time series prediction, but it is not suitable for publication in its current state due to insufficient presentation of the problem definition, inadequate comparative experiments, limitations in performance evaluation, and lack of reproducibility.

In particular, it is essential to conduct a fair comparison with the latest time series models, clarify the hyperparameter optimisation process, introduce various performance metrics and statistical tests, and ensure the interpretability of variables. Additionally, the paper must present its applicability and research contribution in a more persuasive manner.

Therefore, this paper requires a major revision, and it is judged that it will only have publication value if the aforementioned issues are thoroughly addressed.

1. This paper aims to improve performance in specific prediction problems (time series-based), but it is unclear how it specifically fills the gap compared to existing research. It must be clarified whether it simply applies a new model or modifies an existing model, or whether it provides new insights in actual industrial/social applications. The motivation for ‘why this research is necessary’ is lacking. Thank you for your valuable comment. We add in the previous research gaps and necessity of our research in lines 90-96 and 115-135 of introduction.

2. The description of the data set's source, collection process, and preprocessing methods (missing value handling, outlier removal, normalisation, etc.) is insufficient. In particular, in time series prediction research, the periodicity, seasonality, and trend characteristics of the data are important, and the results of exploratory analysis (ACF, PACF, time series decomposition, etc.) should be presented. Otherwise, it is difficult to judge the validity of the modelling. Thank you. There are no missing values in data set. And the outliers were not removed from data sets, because theses outliers are inherent behavior of the ED data set. The ACF and PACF show in Fig 5, and the periodicity and seasonality described in lines 553-563. Also there was no time trend (during years) to decomposition from time series before analysis.

3. The paper mentions hyperparameter settings during model training, but the rationale is insufficient. It should clearly state whether the default values were used or whether methods such as grid search or Bayesian optimisation were applied. The performance of time series prediction models is sensitive to parameters such as learning rate, window size, and hidden dimension, so failure to address these systematically weakens the credibility of the research. Thank you. We describe more in method section lines 281-285. “Extreme Learning Machine (ELM) for time series forecasting is a fast and efficient neural network-based approach that leverages a single-layer feedforward neural network (SLFN) structure. The key idea behind ELM is that the input weights and hidden layer biases are randomly assigned and remain fixed during training, while only the output weights are learned, typically by solving a linear system using matrix pseudo-inversion.

4. Only a few evaluation metrics such as RMSE or MAE were used, which are insufficient to adequately address model characteristics and problem definitions. It is necessary to present various metrics such as MAPE, SMAPE, and R² to verify prediction bias. In addition, statistical significance tests (e.g., Diebold-Mariano test) should be used to prove that the performance differences between models are meaningful. Thank you in advance for this valuable recommendation. We add all these indices in manuscript.

5. The paper focuses solely on improving model performance, but does not analyse which variables (features) contributed significantly to the prediction. In particular, time series prediction research needs to strengthen its explanatory power through variable importance (feature importance) and attention weight visualisation when considering practical application. Simply stating that ‘the accuracy is high’ limits the academic and practical contributions. Thank you. We add the results of ARIMAX in table1, and feature importance of ANN in Fig6. Also more details of variable importance in lines 572-577.

6. The reproducibility is low because the experimental code, parameter details, and hardware/software environment are not specified. In particular, ensuring reproducibility is important in the latest time series model research, but this paper overlooks this point. At the very least, pseudocode, data split method (train/valid/test ratio), and random seed settings should be provided. Thank you. We add all data and codes in supplementary files.

7. In the literature review, recent studies on deep learning models for time series prediction published in the last two to three years were not sufficiently cited. In particular, studies on Transformer-based prediction were omitted, which is a major flaw in clearly establishing the position of this study in academia. Thank you for your suggestion. Our literature review have two aspects: 1- hybride forecastying models 2- models for ED arrivals. We find and read this paper:

“Su, L., Zuo, X., Li, R., Wang, X., Zhao, H., & Huang, B. (2025). A systematic review for transformer-based long-term series forecasting. Artificial Intelligence Review, 58(3), 80.”

So, we find that this models are not popular in ED arrival forecasting.

8. Lack of discussion on how the model presented in the paper can be applied in actual industries, policies, or services. It is difficult to judge the application value based solely on the result that ‘prediction accuracy has improved.’ Thank you. This concerns added in lines 652-672 of discussion.

Reviewer #2:

This manuscript addresses a highly relevant and practice-oriented issue: forecasting daily patient arrivals in emergency departments (EDs) by combining time series and machine learning techniques. The integration of calendar and meteorological variables with both ARIMAX and ANN, along with the development of two hybrid algorithms, is a valuable contribution to healthcare operations research. The comparative evaluation across multiple horizons (short, intermediate, and long term) further enhances the practical applicability of the findings. That said, several conceptual and methodological issues should be addressed to maximize the clarity and impact of the study. The manuscript would benefit from a more structured framing of the literature review, stronger justification for benchmark model selection, clearer reporting of ANN architecture and hyperparameter optimization, and deeper interpretation of horizon-specific performance differences. Additionally, highlighting the unique contributions of this study—beyond confirming the established role of calendar and meteorological factors—would strengthen its significance.

- The manuscript cites a large number of related studies in the introduction; however, these references are presented in a rather descriptive manner, without offering a coherent narrative. Summarizing the common implications and insights of prior research, and explicitly linking their limitations to the distinctive contributions of this study, would make the introduction more structured and persuasive.

- The study proposes two hybrid models and compares them with ARIMA and ANN. However, alternative hybrid models proposed in previous studies, such as the MA-filter based hybrid ARIMA–ANN algorithm (Babu et al., 2014) and the fuzzy time series algorithm (Jilani et al., 2019), were not included as benchmarks. The rationale for excluding these models should be clarified in more detail. Thank you for your valuable comment. We add in the previous research gaps and necessity of our research in lines 90-96 and 115-135 of introduction. And revised the literature review accordingly.

- The manuscript mentions that 40 hidden nodes were used in the ANN model, but it is unclear whether the number of hidden layers was fixed at one. If a single hidden layer was adopted, was this the result of hyperparameter optimization or the authors’ discretion? Furthermore, discussing how ANN performance changes with different configurations of hidden layers and nodes would strengthen the robustness of the modeling approach. Thank you. We describe more in method section lines 281-285. “Extreme Learning Machine (ELM) for time series forecasting is a fast and efficient neural network-based approach that leverages a single-layer feedforward neural network (SLFN) structure. The key idea behind ELM is that the input weights and hidden layer biases are randomly assigned and remain fixed during training, while only the output weights are learned, typically by solving a linear system using matrix pseudo-inversion.

- Figure 6 shows that the variable importance differs across ARIMAX and ANN, even when the same variables are used. An explanation of whether this difference stems from the contrast between linear and nonlinear approaches, or from other factors, would enhance the interpretability of the results. Thank you. We add the results of ARIMAX in table1, and feature importance of ANN in Fig6. Also more details of variable importance in lines 572-577.

- Typically, ANN analyses utilize a broad set of input variables to leverage its predictive capacity. In contrast, this study employed the Maximal Information Coefficient (MIC) approach for variable selection. Providing comparative evidence of model performance with and without the MIC approach would make the methodological contribution clearer. Thank you. The feature selection by MIC were done in ARIMAX not ANN. All input features included in ANN.

- It remains unclear how the dataset was divided into training, validation, and testing sets, and whether repeated experiments were conducted to ensure robustness. Since data splitting strategies and hyperparameter tuning can significantly affect model performance, Table 2 alone may not provide sufficient evidence to conclude that one model is superior to another. Thank you. In lines 172-177 we add details of test and train data partitioning: Model training utilized the initial 1008 days (equivalent to 144 weeks) of available data. To evaluate the robustness of algorithms and forecasting accuracies, the remaining 84 days (12 weeks) were held out for testing. This standardized train/test partition was maintained uniformly across the evaluation of all algorithms: Artificial Neural Network (ANN), ARIMAX, proposed Hybrid models, Long Short-Term Memory (LSTM), and Generalized Linear Model (GLM). Also, we addressed all these concerns in the manuscript.

- The results show that the best-performing model differs by horizon: Hybrid1 performs best in the short horizon, whereas Hybrid2 performs better in the intermediate horizon. The reasons for these differences should be discussed in more depth. Additionally, from a practitioner’s perspective, guidance is needed on which model to use for short-, medium-, and long-term scheduling. If different models are to be applied depending on the time horizon, recommendations on how to integrate their outputs would be valuable. Thank you. It is recommended that the Hybrid 2 is reasonable at all horizons.

- The finding that calendar and meteorological variables significantly influence patient arrivals is consistent with prior literature and may even be considered self-evident. The conclusion would be strengthened by emphasizing what is newly discovered in this study or highlighting its unique contributions beyond confirming established results. Thank you. This concerns added in lines 652-672 of discussion, and in lines 729-735 of conclusion.

---

## [Decision Letter · Decision Letter 1]

19 Feb 2026

PONE-D-25-37594R1Enhancing the Forecast Accuracy of the Daily Number of Patients Arrivals in Emergency Department by Hybrid ARIMAX-ANN AlgorithmPLOS One

Dear Dr. Saki,

Thank you for submitting your manuscript to PLOS ONE. After careful consideration, we feel that it has merit but does not fully meet PLOS ONE’s publication criteria as it currently stands. Therefore, we invite you to submit a revised version of the manuscript that addresses the points raised during the review process.

Especially, please revise the manuscript in light of the reviewer's comments regarding the clarification of research contribution and novelty, fairness in model comparison (experimental design), and the overstatement of the role of MIC-based feature selection.

If applicable, we recommend that you deposit your laboratory protocols in protocols.io to enhance the reproducibility of your results. Protocols.io assigns your protocol its own identifier (DOI) so that it can be cited independently in the future. For instructions see: https://journals.plos.org/plosone/s/submission-guidelines#loc-laboratory-protocols. Additionally, PLOS ONE offers an option for publishing peer-reviewed Lab Protocol articles, which describe protocols hosted on protocols.io. Read more information on sharing protocols at . Additionally, PLOS ONE offers an option for publishing peer-reviewed Lab Protocol articles, which describe protocols hosted on protocols.io. Read more information on sharing protocols at https://plos.org/protocols?utm_medium=editorial-email&utm_source=authorletters&utm_campaign=protocols..

We look forward to receiving your revised manuscript.

Kind regards,

Youngsang Cho

Academic Editor

PLOS One

Journal Requirements:

Reviewers' comments:

Reviewer's Responses to Questions

**Comments to the Author**

1. If the authors have adequately addressed your comments raised in a previous round of review and you feel that this manuscript is now acceptable for publication, you may indicate that here to bypass the “Comments to the Author” section, enter your conflict of interest statement in the “Confidential to Editor” section, and submit your "Accept" recommendation.

Reviewer #1: All comments have been addressed

Reviewer #3: All comments have been addressed

2. Is the manuscript technically sound, and do the data support the conclusions?

Reviewer #1: Yes

Reviewer #3: Yes

3. Has the statistical analysis been performed appropriately and rigorously?

Reviewer #1: Yes

Reviewer #3: No

4. Have the authors made all data underlying the findings in their manuscript fully available?

The PLOS Data policy requires authors to make all data underlying the findings described in their manuscript fully available without restriction, with rare exception (please refer to the Data Availability Statement in the manuscript PDF file). The data should be provided as part of the manuscript or its supporting information, or deposited to a public repository. For example, in addition to summary statistics, the data points behind means, medians and variance measures should be available. If there are restrictions on publicly sharing data—e.g. participant privacy or use of data from a third party—those must be specified. requires authors to make all data underlying the findings described in their manuscript fully available without restriction, with rare exception (please refer to the Data Availability Statement in the manuscript PDF file). The data should be provided as part of the manuscript or its supporting information, or deposited to a public repository. For example, in addition to summary statistics, the data points behind means, medians and variance measures should be available. If there are restrictions on publicly sharing data—e.g. participant privacy or use of data from a third party—those must be specified.

Reviewer #1: Yes

Reviewer #3: Yes

5. Is the manuscript presented in an intelligible fashion and written in standard English?

Reviewer #1: Yes

Reviewer #3: Yes

6. Review Comments to the Author

Reviewer #1: The authors have appropriately addressed and revised the comments. There are no further questions or requests for revisions.

Reviewer #3: The manuscript proposes two hybrid prediction algorithms combining ARIMAX and ANN to forecast daily Emergency Department (ED) visits. Since the first round of review, the authors have made significant efforts to address the comments by enhancing data descriptions, expanding performance metrics, and adding the Diebold–Mariano (DM) test. The overall quality and reproducibility of the manuscript have noticeably improved.

However, from the perspective of a second-round review, the study still faces fundamental weaknesses regarding methodological novelty, the fairness of the comparative experiments, and the theoretical justification for the hybrid structures. While the manuscript is approaching the "technical soundness" criteria of PLOS ONE, further refinement is necessary before a final decision can be made. I recommend a Major Revision (borderline between major and minor).

For more detail, Please confirm the attached file.

7. PLOS authors have the option to publish the peer review history of their article (what does this mean?). If published, this will include your full peer review and any attached files.). If published, this will include your full peer review and any attached files.

**Do you want your identity to be public for this peer review?** For information about this choice, including consent withdrawal, please see our  For information about this choice, including consent withdrawal, please see our Privacy Policy..

Reviewer #1: No

Reviewer #3: No

---

## [Author Response · Author response to Decision Letter 2]

22 Feb 2026

Manuscript ID: PONE-D-25-37594R1

Dear Academic Editor of PLOS one,

Thank you for the opportunity to revise our work. We appreciate the valuable feedback from both reviewers and believe that addressing these comments will significantly improve the manuscript. We highlighted in blue all the revisions. And here is the point-by-point response to the reviewer comments.

Thanks a lot, and best regards,

Azadeh Saki (CA),

Associate Professor of Biostatistics, Department of Epidemiology and Biostatistics, School of Health, Mashhad University of Medical Sciences, Mashhad, Iran,

Mobile : +989163014565

E-mail : Sakia@mum.ac.ir; azadehsaki@yahoo.com

Reviewer comments:

The manuscript proposes two hybrid prediction algorithms combining ARIMAX and ANN to forecast daily Emergency Department (ED) visits. Since the first round of review, the authors have made significant efforts to address the comments by enhancing data descriptions, expanding performance metrics, and adding the Diebold–Mariano (DM) test. The overall quality and reproducibility of the manuscript have noticeably improved.

However, from the perspective of a second-round review, the study still faces fundamental weaknesses regarding methodological novelty, the fairness of the comparative experiments, and the theoretical justification for the hybrid structures. While the manuscript is approaching the "technical soundness" criteria of PLOS ONE, further refinement is necessary before a final decision can be made. I recommend a Major Revision (borderline between major and minor).

1. Clarification of Research Contribution and Novelty

- The authors repeatedly emphasize "two novel hybrid algorithms" as a primary contribution. However, both Hybrid 1 (residual modeling) and Hybrid 2 (simple averaging) appear to be minor variations or standard combinations found in existing ARIMA–ANN literature. Specifically, Hybrid 2 is a basic form of ensemble forecasting, which makes the term "novel algorithm" feel somewhat overstated. The true contribution of this paper lies in its empirical application:

• Testing these models in a specific high-variability context (Mashhad, Iran).

• Incorporating exogenous variables through MIC-based selection.

• Demonstrating how these combinations stabilize predictions across intermediate horizons.

- Tone down the use of "novel algorithm" in the Introduction and Conclusion. Instead, explicitly frame the work as a context-specific empirical study that provides practical insights into ED management.

Thank you for this advance recommendation. We improve the manuscript accordingly in introduction lines 104-125, and discussion section lines: 651-666.

2. Fairness in Model Comparison (Experimental Design)

- While the authors added LSTM and GLM as requested, the setup for LSTM seems insufficient compared to the proposed models. The LSTM is presented with a single fixed setting (lookback=30, epoch=50, batch=30) without mention of hyperparameter tuning or repeated trials. In contrast, the ARIMAX and ANN (ELM) components underwent detailed pre-analysis, such as MIC and FDR. This creates an asymmetric level of optimization, making it difficult to determine whether the hybrid model's superiority over LSTM is due to architectural advantages or simply a lack of tuning for the LSTM baseline.

- At a minimum, clarify that the LSTM serves as an exploratory baseline. Ideally, provide a brief sensitivity analysis or explain the range of tuning used for the comparative models.

Thank you for your valuable comment. We improve the manuscript accordingly in introduction, lines: 139-143, and methods section, lines: 397-403.

3. Overstating the Role of MIC-based Feature Selection

- MIC-based feature selection was applied to ARIMAX and GLM, but ANN and LSTM reportedly received all variables. Despite this, the manuscript often attributes the hybrid model's success primarily to MIC. It remains unclear whether the performance gain in Hybrid 2 stems from:

• The MIC-based selection process.

• The bias-offsetting effect between ARIMAX and ANN.

• The statistical stabilization inherent in simple averaging.

- Attribute the role of MIC specifically to "improving model interpretability and ARIMAX stability" rather than framing it as the sole driver of hybrid performance.

Thank you for your attention. We revise the manuscript accordingly in discussion section, lines: 676-691.

---

## [Decision Letter · Decision Letter 2]

8 Apr 2026

Enhancing the Forecast Accuracy of the Daily Number of Patients Arrivals in Emergency Department by Hybrid ARIMAX-ANN Algorithm

PONE-D-25-37594R2

Dear Dr. Saki,

We’re pleased to inform you that your manuscript has been judged scientifically suitable for publication and will be formally accepted for publication once it meets all outstanding technical requirements.

An invoice will be generated when your article is formally accepted. Please note, if your institution has a publishing partnership with PLOS and your article meets the relevant criteria, all or part of your publication costs will be covered. Please make sure your user information is up-to-date by logging into Editorial Manager at Editorial Manager® and clicking the ‘Update My Information' link at the top of the page. For questions related to billing, please contact  and clicking the ‘Update My Information' link at the top of the page. For questions related to billing, please contact billing support..

Kind regards,

Youngsang Cho

Academic Editor

PLOS One

Additional Editor Comments (optional): Please assign a number to each equation and make sure to cite them in the corresponding paragraphs.

---

## [Editor Report · Acceptance letter]

PONE-D-25-37594R2

PLOS One

Dear Dr. Saki,

I'm pleased to inform you that your manuscript has been deemed suitable for publication in PLOS One. Congratulations! Your manuscript is now being handed over to our production team.

Kind regards,

on behalf of

Prof. Youngsang Cho

Academic Editor

PLOS One